# Metabolic regulation of pluripotency and germ cell fate through α-ketoglutarate

Julia Tischler[1] , Wolfram H Gruhn[1,†], John Reid[2,3,†], Edward Allgeyer[1] , Florian Buettner[4] ,
Carsten Marr[4], Fabian Theis[4,5] , Ben D Simons[1], Lorenz Wernisch[2] & M Azim Surani[1,*]

## Abstract

An intricate link is becoming apparent between metabolism and cellular identities. Here, we explore the basis for such a link in an *in vitro* model for early mouse embryonic development: from naïve pluripotency to the specification of primordial germ cells (PGCs). Using single-cell RNA-seq with statistical modelling and modulation of energy metabolism, we demonstrate a functional role for oxidative mitochondrial metabolism in naïve pluripotency. We link mitochondrial tricarboxylic acid cycle activity to IDH2-mediated production of alpha-ketoglutarate and through it, the activity of key epigenetic regulators. Accordingly, this metabolite has a role in the maintenance of naïve pluripotency as well as in PGC differentiation, likely through preserving a particular histone methylation status underlying the transient state of developmental competence for the PGC fate. We reveal a link between energy metabolism and epigenetic control of cell state transitions during a developmental trajectory towards germ cell specification, and establish a paradigm for stabilizing fleeting cellular states through metabolic modulation.

**Keywords** cell state transitions; germ cells; metabolism; pseudotime analysis; single-cell analysis

**Subject Categories** Development & Differentiation; Metabolism; Stem Cells

The EMBO Journal (2019) 38: e99518

See also: **V Lu & MA Teitell** (January 2019)

## Introduction

Embryonic stem cells (ESCs) have the capacity for indefinite self-renewal *in vitro,* while retaining the ability to differentiate into specialized cell types (Ng & Surani, 2011; Young, 2011). The *in vitro* differentiation of mouse ESCs (mESCs) from a naïve pluripotent state into primed epiblast-like cells (EpiLCs) confers transient developmental competence for the primordial germ cell (PGC) fate (Hayashi *et al*, 2011) and provides a tractable model system for investigations on early embryonic cell state conversions (Fig 1A). These cell states and their transitions are associated with functional heterogeneity, which needs consideration (Cahan & Daley, 2013). While PGCs, the precursors of oocytes and sperm, and naïve ESCs share expression of some key pluripotency transcription factors, together with DNA and histone methylation status, these are distinct cell states (Saitou *et al*, 2003; Seki *et al*, 2005; Surani *et al*, 2007; Hackett & Surani, 2013; Kurimoto *et al*, 2015).

Nutritional state, metabolism and the accompanying epigenetic changes have an impact on cellular identity. For example, threonine metabolism is linked to the synthesis of the methyl donor S-adenosyl-methionine (SAM), which impacts on the histone methylation status, and, in turn, mESC pluripotency (Shyh-Chang *et al*, 2013). Likewise, the metabolite alpha-ketoglutarate (αKG) has a role in mESC self-renewal through enhancing the efficiency of αKG-dependent dioxygenases with key functions in the regulation of epigenetic state (Carey *et al*, 2015), but also in the differentiation of human ESCs (hESCs; TeSlaa *et al*, 2016). Similarly, aerobic glycolysis has been linked to chromatin structure and the maintenance of hESC pluripotency, with glycolysis-derived cytosolic acetyl-CoA serving as an essential substrate for histone acetylation (Moussaieff *et al*, 2015). While primed hESCs depend primarily on aerobic glycolysis, as is the case for the mouse epiblast stem cells (EpiSCs), naïve hESCs and mESCs utilize both glycolysis and oxidative phosphorylation pathways on demand (Zhou *et al*, 2012; Sperber *et al*, 2015). Consistently with their predominantly glycolytic metabolism, stimulating aerobic glycolysis via stabilization of hypoxia-inducible factor 1 alpha (HIF-1α) is sufficient to drive mESCs into epiblast-like cell fates (Zhou *et al*, 2012). Accordingly, activation of oxidative metabolism facilitates the re-acquisition of naïve pluripotency from highly glycolytic EpiSCs (Sone *et al*, 2017), suggesting that changes in cellular metabolism influence cell state transitions. The precise molecular regulation underlying the impact of energy metabolism on mESC pluripotency and during early embryonic development, however, remains poorly defined.

1 Wellcome Trust/Cancer Research UK Gurdon Institute, University of Cambridge, Cambridge, UK
2 MRC Biostatistics Unit, Cambridge Institute of Public Health, University of Cambridge, Cambridge Biomedical Campus, Cambridge, UK
3 The Alan Turing Institute, British Library, London, UK
4 Institute of Computational Biology, Helmholtz Zentrum München–German Research Center for Environmental Health, Neuherberg, Germany
5 Department of Mathematics, Chair of Mathematical Modeling of Biological Systems, Technische Universität München, Garching, Germany
 *Corresponding author. Tel: +44 1223 334136; E-mail: a.surani@gurdon.cam.ac.uk
 †These authors contributed equally to this work

Here, we identify metabolic regulatory pathways that are dynamically modulated during the conversion from naïve to primed pluripotency in mouse, and establish the influence of oxidative metabolism on mESC pluripotency and developmental competence for the PGC fate. We link oxidative mitochondrial metabolism and tricarboxylic acid cycle to the production of αKG and, in turn, the activity of key epigenetic regulators. On the basis of our findings, we propose a metabolic regulatory mechanism via αKG, which mediates early embryonic cell state transitions and germ cell development through promoting permissive epigenetic states.

## Results

### Single-cell analysis reveals metabolic regulatory dynamics and competence for the PGC fate

First, we used the *in vitro* differentiation of naïve mouse embryonic stem cells (ESCs) from pluripotent ground state (2i/Lif culture conditions; Ying *et al*, 2008) into primed epiblast-like cell (EpiLC)

fates (Hayashi *et al*, 2011; Fig 1A), and performed single-cell RNA-sequencing (RNA-seq) at $t = 0$, $t = 24$ and $t = 48$ h (Fig EV1A and B). Gaussian process latent variable models (GPLVMs), a non-linear dimensionality reduction approach (Lawrence, 2004; Buettner & Theis, 2012), grouped individual cells into distinct transcriptional states, which were highly correlated with sampling time (Fig EV1C). We harnessed the cellular heterogeneity arising during EpiLC differentiation to derive dynamic gene expression trajectories by statistically ordering single-cell transcriptomes over a developmental time ("pseudotime"; Trapnell *et al*, 2014; Reid & Wernisch, 2016; Figs 1B and EV1D), and comprehensively quantified expression level changes (Appendix Table S1). Key regulators of naïve pluripotency, such as *Esrrb* and *Tfcp2l1*, displayed pronounced downward pseudotime profiles, while genes associated with epiblast development, such as *Fgf5* and *Lin28b*, showed increasing expression over time; this recapitulates the known expression dynamics (Hayashi *et al*, 2011). Central regulators of energy metabolism exhibited similarly dynamic trajectories. Accordingly, pyruvate-dependent kinases 1 and 3 (*Pdk1* and *Pdk3*) and *Slc2a1* and *Stk11* (Fig EV1D) were upregulated over time, conceivably contributing to enhanced glycolysis by suppressing entry

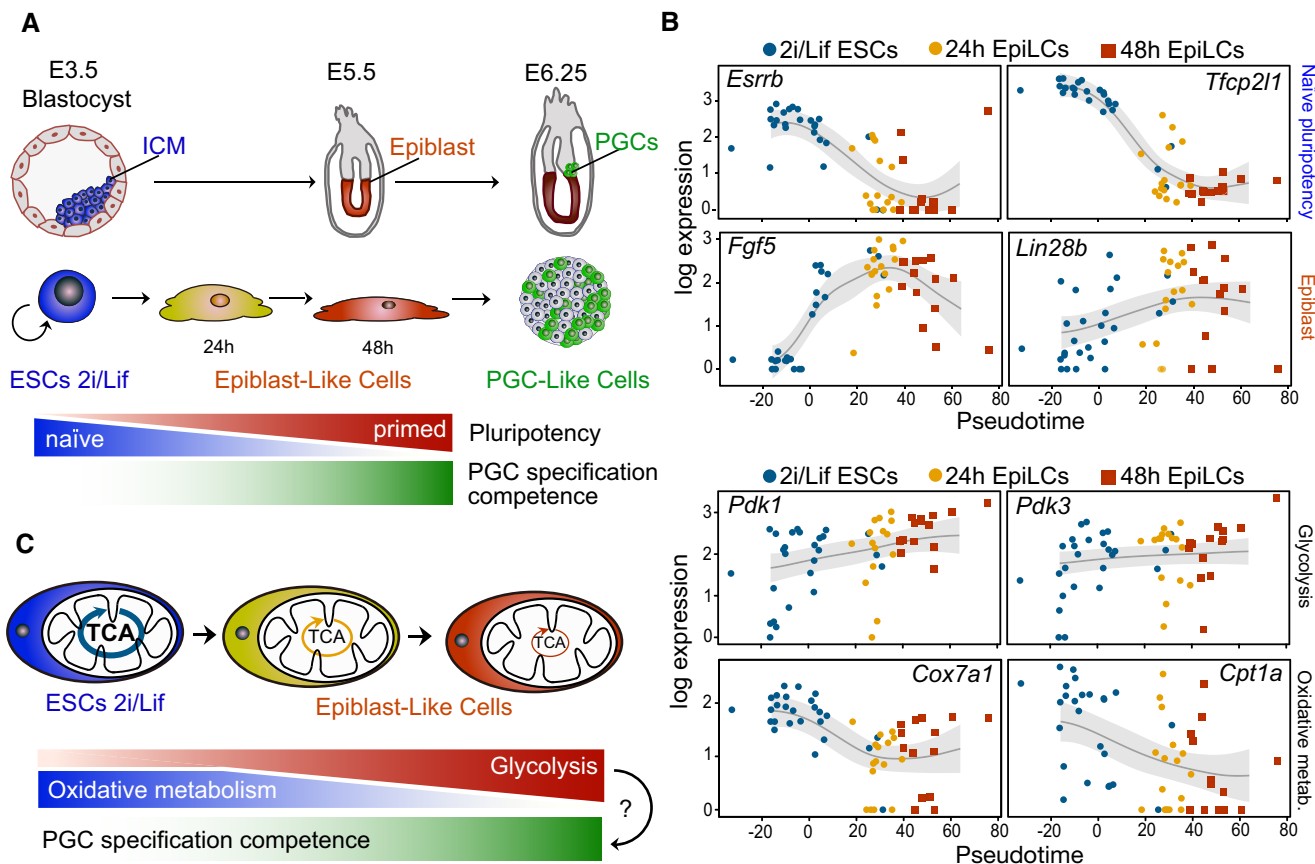

**Figure 1. Embryonic cell state transitions underlie dynamic changes in energy metabolism.**

A    Model depicting the conversion of mouse embryonic stem cells (ESCs) from a naïve pluripotent state in 2i/Lif culture conditions to primed epiblast-like cells (EpiLCs), which acquire transient competence for induction into primordial germ cell (PGC)-like cell fate. Corresponding developmental stages are shown in the mouse embryo.

B    Single-cell expression data in pseudotime of central regulators of naïve pluripotency, epiblast marker genes, glycolytic regulators and genes with key functions in oxidative metabolism.

C    Schematic illustrating the dynamic changes in energy metabolism during the acquisition of developmental competence for the PGC fate. TCA, tricarboxylic acid cycle.

Data information: See also Fig EV1.

of pyruvate into the mitochondrial tricarboxylic acid (TCA) cycle and by facilitating glucose uptake, respectively. Conversely, genes with central roles in oxidative metabolism, such as *Cox7a1* and *Cpt1a*, exhibited a prominent decline. These dynamic expression changes suggest a switch to an increased glycolytic state with a concomitant decrease in oxidative metabolism (Fig 1C), as cells acquire competence for the PGC fate (Zhou *et al*, 2012; Zhang *et al*, 2016).

## Oxidative mitochondrial metabolism maintains an embryonic stem cell-like state

Next, we investigated potential implications of sustained oxidative mitochondrial metabolism through repression of glycolysis for naïve

pluripotency (Fig 2A). Using a knock-in reporter ESC line expressing a destabilized green fluorescent protein from the endogenous *Zfp42/Rex1* locus (*Rex1*-GFPd2; Wray *et al*, 2011; Kalkan *et al*, 2017), we found that inhibition of glycolysis through supplementation of the glucose analogue 2-deoxy-D-glucose (2-DG; Wick *et al*, 1957; Zhou *et al*, 2012) prevented the exit from naïve pluripotency, indicated by the sustained expression of *Rex1*-GFPd2, in a dose-dependent manner (Figs 2B and EV2A–C). Expression levels of marker genes for naïve pluripotency, including *Esrrb*, *Klf4* and *Tfcp2l1*, that were strongly downregulated in controls by $t = 48$ h during the ESC-to-EpiLC transition remained elevated following 2-DG treatment (Fig 2C). Conversely, epiblast markers, such as the *de novo* methyltransferase *Dnmt3b*, *Fgf5* and *Lin28b*, were

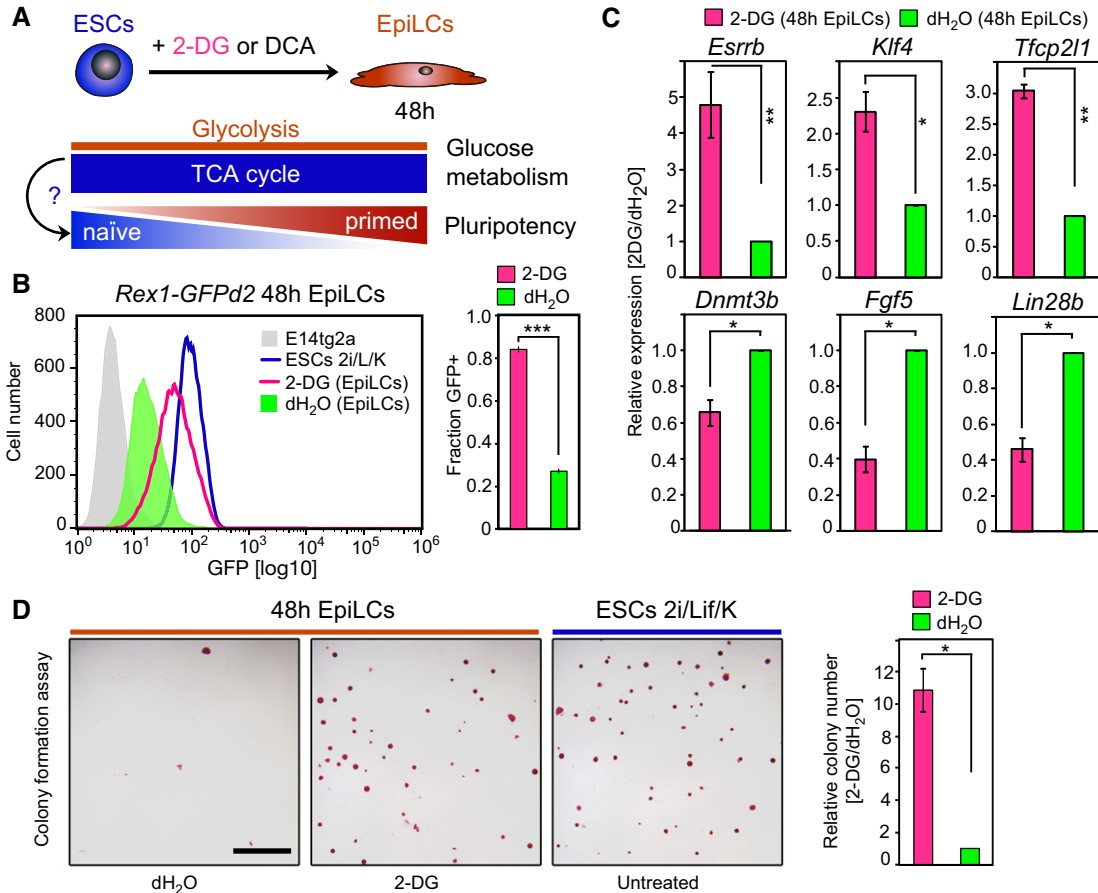

**Figure 2.  Oxidative metabolism promotes naïve pluripotency.**

A   Investigating the effect of sustained oxidative mitochondrial metabolism on the transition from naïve ESCs to primed EpiLCs through supplementation of the glycolysis inhibitors 2-deoxy-D-glucose (2-DG) and dichloroacetate (DCA), respectively.

B   Flow cytometry analysis of *Rex1*-GFPd2 cells following addition of 10 mM 2-DG during the 48 h EpiLC induction. Representative GFP intensity distributions are depicted. Average proportions of *Rex1*-GFPd2-positive (GFP+) cells are quantified from two independent biological replicates. Error bars represent ± SE. ***$P$ = 0.0006 (unpaired 1-tailed Student's *t*-test).

C   Expression analysis by qRT–PCR of naïve pluripotency and epiblast marker genes in bulk 48 h cells after 10 mM 2-DG treatment. Relative expression levels, normalized to control culture conditions, are shown. Graphs represent averages from triplicate (duplicate for *Klf4* and *Tfcp2l1*) independent biological experiments. Error bars denote ± SE. *$P$ ≤ 0.05; **$P$ ≤ 0.01 (unpaired 1-tailed Student's *t*-test, see Appendix Table S3 for exact $P$-values).

D   Colony-forming ability following 10 mM 2-DG supplementation during EpiLC stimulation. Representative images of alkaline phosphatase (AP)-stained colonies are displayed. Scale bar, 250 μm. The average colony formation, normalized to control culture conditions, quantified from two independent biological replicates, is shown. Error bars signify ± SE. *$P$ = 0.0424 (unpaired 1-tailed Student's *t*-test).

Data information: See also Fig EV2.

**Figure 3. αKG maintains naïve pluripotency.**

A Pseudotime expression profiles for the αKG-regulating enzymes *Idh2* and *Dlst* during the transition from naïve to primed pluripotency. TCA cycle enzymes and metabolites produced within the TCA cycle are illustrated.

B Representative flow cytometry profiles of *Rex1*-GFPd2 cells following 4 mM dm-αKG supplementation during the EpiLC induction are depicted. Graphs show average fractions of *Rex1*-GFPd2-positive (GFP+) cells from six independent biological assays. Error bars indicate ± SE. ***$P$ = 1.241E-05 (unpaired 1-tailed Student's *t*-test).

C qRT–PCR analysis of naïve pluripotency regulators and epiblast marker genes following EpiLC stimulation in the presence of 4 mM dm-αKG. Expression data are normalized to control culture conditions and represent averages from five biological replicates in bulk 48 h cells. Error bars denote ± SE. ***$P \leq 0.005$ (unpaired 1-tailed Student's *t*-test, see Appendix Table S3 for exact $P$-values).

D Colony-forming ability succeeding 4 mM dm-αKG treatment during the 48 h EpiLC induction. Characteristic images of AP-stained colonies are shown. Scale bar, 250 μm. Colony formation is normalized to control-treated cells and quantified from quadruplicate experiments. Error bars signify ± SE. *$P$ = 0.0283 (unpaired 1-tailed Student's *t*-test).

E Representative super-resolution images of TOM-20 immune-labelled mitochondria in ESCs following 48 h culture in 2i/Lif/KSR media, and EpiLC-inducing conditions in the presence of 4 mM dm-αKG and DMSO, respectively, are displayed. Scale bar, 3 μm.

F–H Ten-day culture of *Rex1*-GFPd2 cells in N2B27/Lif/KSR with 4 mM dm-αKG and DMSO, respectively, with passaging every 2.5 days. (F) Characteristic bright-field images of *Rex1*-GFPd2 cells after 10 days of culture in 2i/Lif/KSR and N2B27/Lif/KSR, in the presence of dm-αKG and DMSO, respectively. Scale bar, 10 μm. (G) Flow cytometer-based quantification of *Rex1*-GFPd2-positive (GFP+) cells. Representative GFP intensity distributions are displayed. The average fractions of GFP+ cells are measured from duplicate experiments. Error bars denote ± SE. *$P$ = 0.0477 (unpaired 1-tailed Student's *t*-test). (H) qRT–PCR analysis of naïve pluripotency and differentiation markers in bulk cells harvested at 2.5-day intervals during the 10-day culture in N2B27/Lif/KSR with dm-αKG or DMSO. Expression data are normalized to time-matched ESCs in 2i/Lif/KSR culture conditions and are averaged over two independent biological experiments. Error bars indicate ± SE. *$P \leq 0.05$; **$P \leq 0.01$; ***$P \leq 0.005$ (unpaired 1-tailed Student's *t*-test, see Appendix Table S3 for precise $P$-values).

I Model illustrating the IDH2-mediated production of αKG in the mitochondrial TCA cycle during oxidative metabolism in ESCs in naïve pluripotency conditions.

Data information: See also Figs EV3 and EV4.

repressed (Fig 2C). Further, glycolytic suppression also had an impact on colony-forming ability, a hallmark of naïve pluripotency. While ESCs have the potential to self-renew and can generate colonies from single cells in naïve pluripotency-promoting conditions, this ability is lost in 48 h EpiLCs (Murakami *et al*, 2016). The addition of 2-DG during the ESC-to-EpiLC differentiation, however, resulted in the subsequent robust self-renewal and colony formation (Fig 2D), supporting the maintenance of a naïve pluripotent state. Comparable results were obtained following treatment with the PDK inhibitor dichloroacetate (DCA), which enhances the conversion of pyruvate to acetyl-CoA in the mitochondria (Stacpoole, 1989; Whitehouse *et al*, 1974; Fig EV2D–G).

Collectively, activation of mitochondrial oxidative metabolism through inhibition of glycolysis sustains an ESC-like state, suggesting that the oxidative-to-glycolytic switch might functionally promote the conversion from naïve to primed pluripotency and consequently the acquisition of developmental competence for the PGC fate.

**The TCA cycle metabolite αKG mediates the naïve pluripotency-promoting effect of oxidative mitochondrial metabolism**

Pseudotime expression profiles and quantitative analysis of enzymes central to the mitochondrial TCA cycle revealed pronounced downregulation of the αKG-producing enzyme *Idh2* but slight upregulation of the αKG-to-succinate-converting enzyme *Dlst* (Fig 3A, Appendix Table S1), suggesting that αKG levels are diminished during the transition from naïve to primed pluripotency. Correspondingly, IDH2 protein levels were distinctly lower in 48 and 72 h EpiLCs, as compared to naïve ESCs (Fig EV3A).

To investigate a potential functional link between oxidative mitochondrial metabolism, TCA cycle activity, and αKG levels, we examined the effect of sustained αKG supplementation on the ESC-to-EpiLC transition. Addition of dimethyl-αKG (dm-αKG) during the 48 h EpiLC induction resulted in the retention of *Rex1*-GFPd2-positive cells in a dose-dependent manner (Figs 3B, and EV3B and C).

Indeed, cells cultured in 4 mM dm-αKG (Carey *et al*, 2015) retained a homogeneous *Rex1*-high state, with a GFP intensity distribution resembling naïve ESCs (Fig 3B). As in naïve ESCs, IDH2 levels were high following dm-αKG supplementation during the EpiLC stimulation, which is consistent with an active TCA cycle (Fig EV3A and D). Accordingly, dm-αKG treatment during the EpiLC differentiation promoted expression of the naïve pluripotency regulators *Esrrb*, *Klf4* and *Tfcp2l1*, while *Dnmt3b*, *Fgf5* and *Lin28b*, marker genes of epiblast, remained low (Figs 3C and EV3E). Moreover, the colony-forming ability was strongly enhanced following EpiLC stimulation in the presence of dm-αKG (Figs 3D and EV3F). The functional similarity of cells subsequently to dm-αKG supplementation during the 48 h EpiLC induction to naïve ESCs was also reflected in the mitochondrial morphology. Super-resolution imaging of immune-stained outer mitochondrial membrane protein TOM-20 (Huang *et al*, 2016) showed that, in the presence of dm-αKG, mitochondria maintained a naïve ESC-like oval morphology and did not form elongated shapes, as observed in control EpiLCs (Figs 3E and EV3G).

The effect of dm-αKG was reversible and did not compromise ESC pluripotency and differentiation potential (Fig EV3H). Dm-αKG pre-treatment of naïve ESCs in 2i/Lif culture conditions, however, led to the dose-dependent retention of cells in a *Rex1*-GFPd2-positive state during transition to EpiLCs (Fig EV3I), proposing that the intracellular αKG levels might need to diminish for an exit from naïve pluripotent state.

Together, our molecular, functional and morphological characterizations suggest that αKG sustains an ESC-like state during EpiLC induction.

**The naïve pluripotency-promoting effect is specific to the TCA cycle metabolite αKG**

We then asked whether TCA cycle metabolites other than αKG might support a naïve pluripotent state. Supplementation of citrate,

 

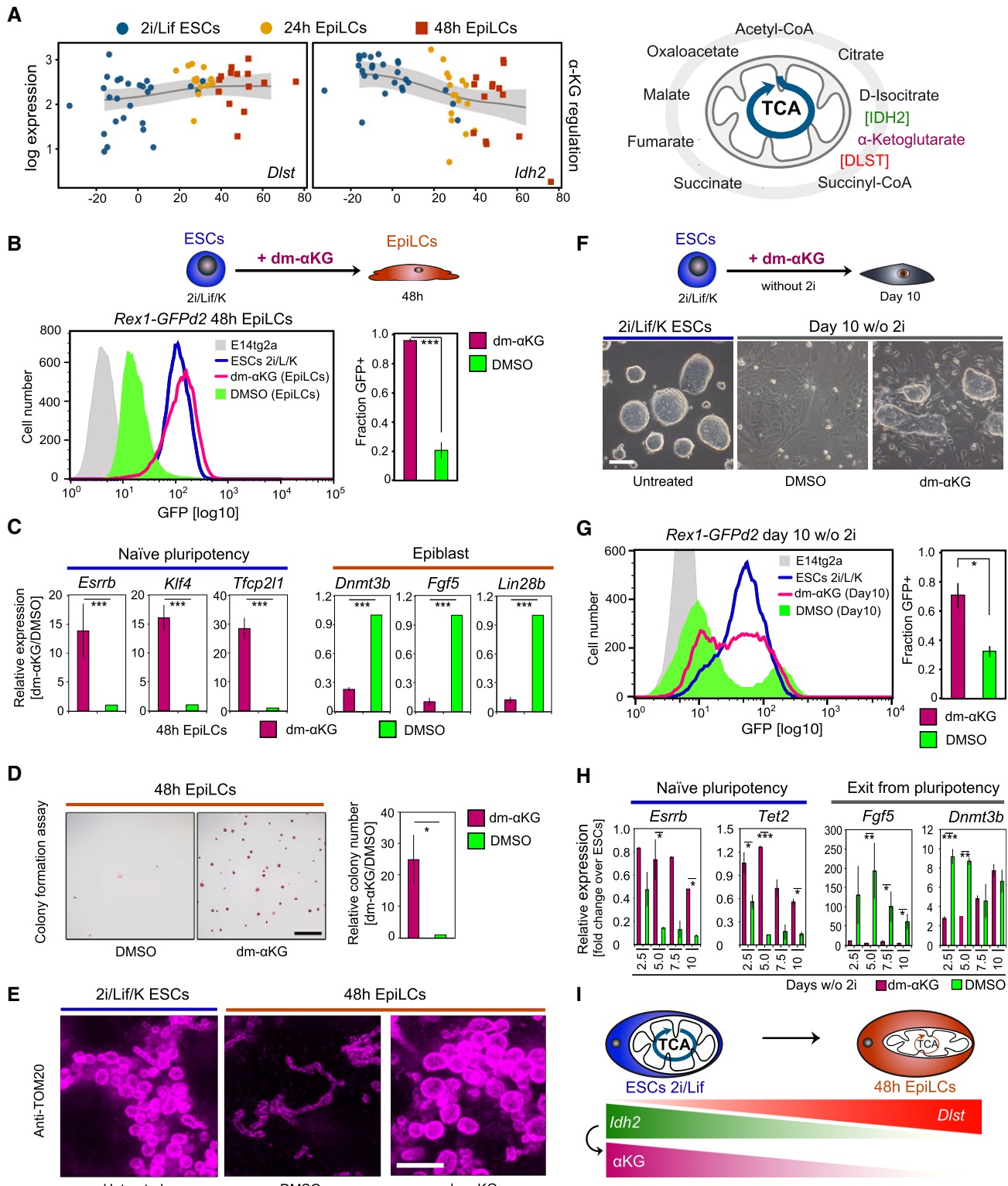

**Figure 3.**

a key metabolite upstream of αKG in the mitochondrial TCA cycle (Fig 3A), during the EpiLC induction resulted in a moderate increase in the fraction of *Rex1*-GFPd2-positive cells (Fig EV4A and B) and colony-forming ability (Fig EV4C). Addition of the downstream metabolite succinate, however, led to the loss of *Rex1*-GFPd2 expression and colony formation, comparable to control EpiLCs (Fig EV4A–C). Together, these results suggest a highly specific function for αKG in the maintenance of an ESC-like state. We thus propose that αKG relays the effect of enhanced oxidative metabolism and TCA cycle activity on naïve pluripotency.

## αKG promotes naïve pluripotency over multiple passages in the absence of 2i inhibitors

Next, we explored whether αKG can replace 2i inhibitors in sustaining naïve pluripotency. Addition of dm-αKG to N2B27 media supplemented with Lif and KSR, which on its own rapidly induces differentiation, supported round, dome-shaped colony morphology, similar to naïve ESCs, over at least 10 days and four passages, respectively (Fig 3F). A minor fraction of cells differentiated in the presence of dm-αKG, as judged by their elongated, flat shape, reminiscent of feeder cells forming a support layer. Flow cytometry analysis of *Rex1*-GFPd2 reporter cells confirmed observations from visual inspection; following 10 days of culture with dm-αKG, a large proportion of cells were *Rex1*-GFPd2-positive (Fig 3G). Transcript levels of the ESC marker genes *Esrrb* and *Tet2* remained elevated in the presence of dm-αKG, further supporting maintenance of naïve pluripotency (Fig 3H). Together, these data suggest that αKG can, at least partially, replace 2i inhibitors in the culture media to sustain an ESC-like state over multiple passages.

## αKG supports naïve pluripotency via cell cycle-dependent and independent mechanisms

We then asked whether the effect of αKG was due to a decrease in cellular proliferation (Fig EV4D). We thus assessed whether the naïve pluripotency-promoting effect specific to dm-αKG was conferred through its direct impact on proliferation, or whether it was mediated primarily via cell cycle-independent mechanisms. Slowing down proliferation rates by treatment with a cyclin-dependent kinase 4 (CDK4) cell cycle inhibitor (CDK4i; Zhu *et al*, 2003; Roccio *et al*, 2013) during the ESC-to-EpiLC transition led to the dose-dependent retention of cells in a *Rex1*-GFPd2-positive state (Fig EV4E), demonstrating that slowing down of the cell cycle delayed exit from naïve pluripotency. However, at equivalent proliferation rates, the fraction of *Rex1*-GFPd2-positive cells following dm-αKG supplementation exceeded the fraction of *Rex1*-GFPd2-positive cells following CDK4 inhibition (Fig EV4F). Furthermore, at matching proliferation rates, expression levels of marker genes for naïve pluripotency were significantly higher in dm-αKG-treated cells, as compared to CDK4-inhibited cells (Fig EV4G). Thus, by revealing an enhanced effect of dm-αKG treatment on naïve pluripotency over merely reduced proliferation rates, our data point to additional, cell cycle-independent effects underlying the impact of αKG on pluripotent state.

## αKG supports ESC pluripotency via maintenance of a naïve epigenetic state

αKG is a known co-factor for a multitude of αKG-dependent dioxygenases, many of which play central roles in the regulation of chromatin structure, such as the histone H3 lysine 9 dimethyl (H3K9me2) demethylases KDM3A and KDM3B, and the ten-eleven translocation (TET) enzymes TET1 and TET2 (Klose *et al*, 2006; Kaelin, 2011; Losman & Kaelin, 2013). Consistently, combinatorial knockdown of the H3K9me2 demethylases *Kdm3a* and *Kdm3b* resulted in the reduced colony formation following EpiLC induction in the presence of dm-αKG (Fig EV4H and I). Accordingly, differences in expression levels of selected ESC and epiblast marker genes were minimized between dm-αKG- and

control-treated EpiLCs in *Tet1/Tet2* double-knockout (DKO; Dawlaty *et al*, 2013) cells (Fig EV4J). This suggests that αKG supports naïve pluripotency, at least in part, through increasing the efficiency of KDM3A and KDM3B, and TET1 and TET2, respectively.

## αKG promotes germ cell fate

Expression of naïve pluripotency genes in primordial germ cells (PGCs), the precursors of sperm and eggs, indicates that they have a role in a different context (Saitou *et al*, 2003). Remarkably, as in naïve ESCs, the genes encoding for COX7A1, a central regulator of mitochondrial oxidative metabolism, and the αKG-producing enzyme IDH2 are upregulated in PGC-like cells (PGCLCs) generated from EpiLCs via embryoid body formation in the presence of cytokines (Hayashi *et al*, 2011; Fig EV5A). This suggests that oxidative metabolism and αKG synthesis are enhanced during PGC development. We also note increased expression of *Pdk1/3* in PGCLCs, which merits further investigation in the future. Thus, to examine the impact of αKG on PGC fate, we induced PGCLCs from *Prdm1*-GFP (Ohinata *et al*, 2005) EpiLCs. PGCLC stimulation under addition of dm-αKG led to a roughly 50% increase in the proportion of *Prdm1*-GFP-positive cells by day 4 (Figs 4A and EV5B), albeit with a slightly reduced PGCLC embryoid size, likely due to αKG's impact on cellular proliferation. The key PGC regulators *Prdm1*, *Prdm14*, *Tfap2c* and *Brachyury* (*T*) were highly expressed, while the ESC-specific gene *Klf4* was repressed in *Prdm1*-GFP-positive PGCLCs induced in the presence of dm-αKG (Fig EV5C). Transcript levels of the endoderm-specific gene *Gata6* were low, suggesting that dm-αKG was specifically enhancing PGC fate. Moreover, robust expression of the αKG-dependent methylcytosine dioxygenase 1, *Tet1*, and the H3K9me2 demethylases *Kdm3a* and *Kdm3b* is noteworthy, as these changes allow for the loss of DNA methylation in PGCs. Collectively, our data suggest that dm-αKG supports specification of *Prdm1*-GFP-positive PGCLCs.

Stimulation with BMP4 alone is sufficient to drive PGC development within 2 days (Aramaki *et al*, 2013). We next explored the impact of dm-αKG without BMP4/8 in inducing PGC fate. Indeed, dm-αKG increased the proportion of *Prdm1*-GFP-positive cells within 2–4 days by almost twofold over controls (Figs 4B, and EV5D and F), with pronounced expression of *Prdm1*, *Prdm14* and *Tfap2c* (Fig EV5E). These data indicate that dm-αKG alone is sufficient to stimulate PGC development from EpiLCs, albeit with reduced efficiency. This increase was partially reversed by treatment with LDN-193189, a small molecule inhibitor of BMP type I receptors (Loh *et al*, 2014; Fig EV5F), suggesting that αKG acts in concert with endogenous BMP signalling to promote PGCLC differentiation.

## αKG safeguards the transient state of developmental competence for the PGC fate

Next, we investigated the impact of αKG on the PGC specification competency. Addition of dm-αKG from 24 to 48 h after the initiation of EpiLC differentiation significantly reduced the number of *Prdm1*-GFP-positive cells in day-4 PGCLC aggregates (Fig EV6A and B), conceivably through retaining cells in an ESC-like state. Dm-αKG

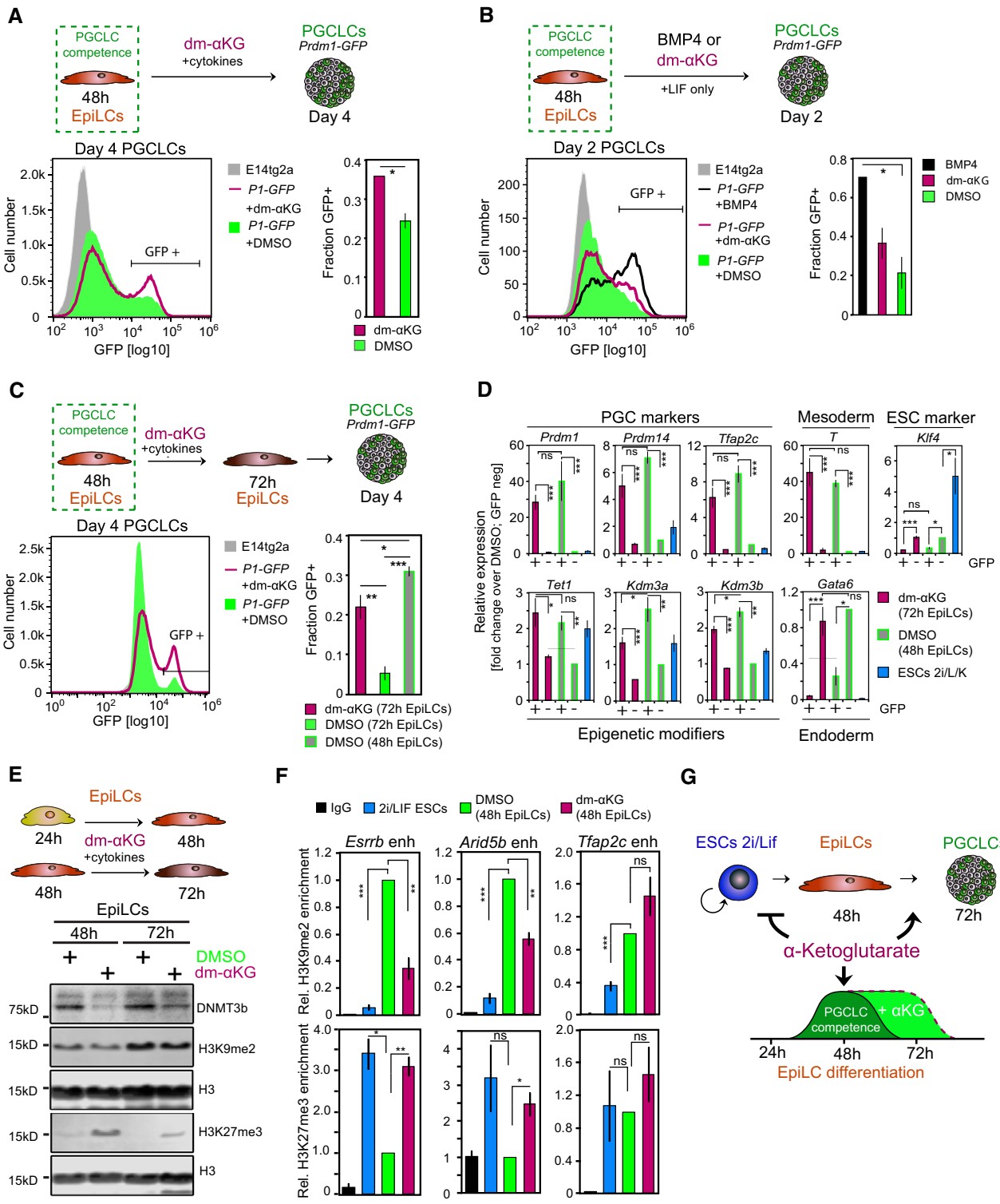

**Figure 4.**

supplementation at 48 h, however, during the course of PGCLC induction, resulted in a robust proportion of *Prdm1*-GFP-expressing cells in day-4 PGCLC embryoids (Figs 4A and EV5B). Remarkably, the addition of dm-αKG at 48 h during the EpiLC differentiation also markedly prolonged the transient state of competence from its peak

at 48 h, up to 72 h (Figs 4C and EV6C). Thus, the induction efficiency of *Prdm1*-GFP-positive PGCLCs from dm-αKG-treated 72 h EpiLCs was comparable to those specified from control-treated 48 h EpiLCs. These cells showed appropriate expression of the key germ cell regulators *Prdm1*, *Prdm14*, *Tfap2c* and *Brachyury* (*T*)

◀

**Figure 4. αKG promotes PGC fate.**

A    FACS analysis of *Prdm1*-GFP-positive (GFP+) cells in day-4 embryoids specified in the presence of 4 mM dm-αKG and PGC cytokines. Representative flow cytometer profiles are depicted. Graphs show the average fractions of GFP+ cells from duplicate experiments. Error bars denote ± SE. *$P = 0.0526$ (unpaired 1-tailed Student's *t*-test). P1-GFP, *Prdm1*-GFP.

B    FACS analysis of *Prdm1*-GFP-positive (GFP+) cells in day-2 embryoids aggregated under addition of LIF (10 ng ml$^{-1}$) and BMP4 (500 ng ml$^{-1}$), dm-αKG (4 mM) or DMSO. Representative flow cytometer profiles are displayed. Average fractions of GFP+ cells are calculated from duplicate assays. Error bars denote ± SE. *$P = 0.0526$. P1-GFP, *Prdm1*-GFP.

C    FACS analysis of *Prdm1*-GFP-positive (GFP+) cells in day-4 PGCLC aggregates specified from 4 mM dm-αKG-treated ($t = 48$ to $t = 72$ h) EpiLCs. Representative flow cytometer profiles are depicted. The average fractions of GFP+ cells, quantified from triplicate experiments, are shown. Error bars signify ± SE. *$P ≤ 0.05$; **$P ≤ 0.01$; ***$P ≤ 0.005$ (unpaired 1-tailed Student's *t*-test, see Appendix Table S3 for precise *P*-values). P1-GFP, *Prdm1*-GFP.

D    Transcript analysis by qRT–PCR of PGC specifiers, demethylating enzymes, mesoderm, endoderm and ESC regulators in FACS-sorted day-4 *Prdm1*-GFP embryoids induced from 4 mM dm-αKG-treated 72 h EpiLCs. Expression levels are normalized to *Prdm1*-GFP-negative cells from control embryoids. Graphs represent averages from triplicate experiments. Error bars indicate ± SE. *$P ≤ 0.05$; **$P ≤ 0.01$; ***$P ≤ 0.005$ (unpaired 1-tailed Student's *t*-test, see Appendix Table S3 for precise *P*-values). +, *Prdm1*-GFP-positive cells; −, *Prdm1*-GFP-negative cells.

E    Western blot analysis for H3K9me2, H3K27me3 and DNMT3b in 4 mM dm-αKG-treated EpiLCs. H3 is used as a loading control.

F    ChIP-qPCR analysis of H3K9me2 and H3K27me3 in putative enhancer regions of genes associated with the naïve pluripotent state (*Esrrb*, *Arid5b*) and PGC fate (*Tfap2c*), respectively, in naïve ESCs and at $t = 48$ h following EpiLC induction in the presence of 4 mM dm-αKG and DMSO, respectively. Graphs show enrichment of H3K9me2, H3K27me3 and IgG control, respectively, relative to DMSO-treated EpiLCs. Averages represent triplicate independent experiments. Error bars signify ± SE. *$P ≤ 0.05$; **$P ≤ 0.01$; ***$P ≤ 0.005$ (unpaired 1-tailed Student's *t*-test, see Appendix Table S3 for precise *P*-values).

G    Model illustrating the extension of the transient developmental competence for the PGC fate through αKG.

Data information: See also Figs EV5 and EV6.

(Fig 4D). Accordingly, *Tet1*, *Kdm3a* and *Kdm3b* were robustly expressed, while the ESC-specific regulator *Klf4* and the endoderm-specific marker gene *Gata6* were repressed (Fig 4D). Of note, in the control EpiLCs, the competent state for the specification of *Prdm1*-GFP-positive PGCLCs largely declines after 48 h (Figs 4C and EV6C).

Acquisition of competence for the PGC fate is associated with dynamic changes in activities of epigenetic regulators (Surani *et al*, 2007; Hayashi *et al*, 2011; Hackett & Surani, 2013; Kurimoto *et al*, 2015), which include those that are modulated by αKG (Klose *et al*, 2006; Kaelin, 2011; Losman & Kaelin, 2013). Global H3K9me2 levels rose and H3K27me3 levels declined during the EpiLC differentiation (Fig EV3A), which recapitulates the known histone methylation dynamics during early mouse development (Kurimoto *et al*, 2015; Zylicz *et al*, 2015). Notably, dm-αKG treatment preceding the acquisition of developmental competence for the PGC fate prevented cells from attaining the high H3K9me2 and, conversely, low H3K27me3 levels detected in the control 48 h EpiLCs (Fig 4E), in line with the characteristic epigenetic state of a relatively naïve cell state (Fig EV3A). Similarly, dm-αKG supplementation beyond the time of acquisition of developmental competence broadly sustained H3K9me2 and H3K27me3 levels of PGC-competent 48 h control EpiLCs (Fig 4E). Likewise, in dm-αKG-treated EpiLCs, the levels of the *de novo* DNA methyltransferase, DNMT3b, were maintained unlike in controls, which showed an increase between 48 and 72 h (Fig 4E). These results suggest that αKG stabilizes the transient developmental potential for the PGC fate through preserving the particular epigenetic state of competent EpiLCs.

Notably, locus-specific analysis of H3K9me2 and H3K27me3 by ChIP-qPCR revealed that dm-αKG counteracts the accumulation of H3K9me2, particularly on some enhancer elements associated with the genes for the naïve pluripotent state (Zylicz *et al*, 2015), such as *Esrrb*, *Nanog*, *Prdm14* and *Arid5b* (Figs 4F and EV6D). Consistently, levels for the H3K27me3 mark were higher on these loci, except for *Prdm14*, where we detected no change. However, on loci, such as in the enhancer region of the PGC regulator *Tfap2c*, both repressive marks are increased. This is in line with a general

repression of germline genes during epiblast development (Kurimoto *et al*, 2015). We reason that this locus-specific effect might reflect the selective recruitment of αKG-dependent H3K9me2 demethylases, which is consistent with the subtle changes in global H3K9me2 levels.

In summary, αKG enhances PGCLC differentiation potential via synergistic action with BMP signalling and prolongs the time of developmental competence for PGC specification, at least partially, through maintaining H3K9me2, H3K27me3 and DNMT3b largely at levels of PGC-competent 48 h EpiLCs. We also uncover a locus-specific effect of dm-αKG; we find that low levels of H3K9me2 mark a subset of cis-regulatory regions, in particular enhancers of pluripotency-associated genes. In contrast, other regulatory regions, such as enhancers of germ cell-associated genes, show an increase in their H3K9me2 levels, irrespective of dm-αKG treatment. Through safeguarding a permissive epigenetic state for the PGC fate, αKG might recruit a larger proportion of cells into the competent state, which, in turn, increases the number of *Prdm1*-GFP-positive PGCLCs. Together, our findings extend the interrelation between an oxidative metabolic state, the central TCA cycle metabolite αKG, methylation status and naïve pluripotency, to germ cell development (Figs 4G and EV6E).

## Discussion

Single-cell RNA-seq during the *in vitro* transition of naïve mouse embryonic stem cells (ESCs) into primordial germ cell (PGC)-competent epiblast-like cells (EpiLCs) and quantitative data analysis support a metabolic switch from an oxidative to an exceedingly glycolytic state (Zhou *et al*, 2012; Zhang *et al*, 2016). Correspondingly, we reveal dynamic upregulation of *Lin28b*, which plays a crucial role in the suppression of genes involved in oxidative metabolism and the regulation of mammalian glucose metabolism (Zhu *et al*, 2011; Zhang *et al*, 2016). Pseudotime trajectories also recapitulate other known metabolic regulatory dynamics, such as high expression of threonine dehydrogenase (*Tdh*) in naïve ESCs, with a sharp decline during EpiLC differentiation (Fig EV1D,

Appendix Table S1), consistent with the requirement of threonine metabolism for maintaining ESC pluripotency (Shyh-Chang *et al*, 2013). A shift to a predominantly glycolytic metabolism in the post-implantation epiblast (Zhou *et al*, 2012; Zhang *et al*, 2016) seems important for the competent state and PGC fate.

We propose a critical function for an active mitochondrial oxidative metabolism in the replenishing of intracellular αKG levels, which, in turn, promotes demethylating enzymes central to naïve pluripotency and PGC fate. Transcript and protein dynamics of the TCA cycle enzyme IDH2 suggest that αKG levels accumulate in the naïve pluripotent state and decline during EpiLC differentiation (Fig 3I). Consistently, recent studies measured intracellular αKG concentrations to be lower in primed (serum/Lif-cultured) or differentiated cells as compared to naïve ESCs (Carey *et al*, 2015; Hwang *et al*, 2016). However, in contrast to these reports, which link regulation of intracellular αKG production to glycolysis-coupling pathways (Hwang *et al*, 2016) and glutamine metabolism (Carey *et al*, 2015), respectively, our results show negligible changes in expression levels of enzymes implicated in the conversion of αKG from glutamine and glutamate (*Gls*, *Gls2*, *Glul*, *Glud1*) or the glycolysis-branched transaminases *Psat1* and *Psph*, during the ESC-to-EpiLC transition (Fig EV1D, Appendix Table S1). Instead, we propose that enhanced mitochondrial oxidative metabolism and TCA cycle stimulate αKG production from citrate through mitochondrial IDH2 in naïve ESCs. Accordingly, our analysis reveals binding of the key pluripotency factors OCT4, SOX2 and NANOG (OSN) in the promoter region of *Idh2* (Appendix Fig S1). No OSN binding by our criterion is observed in the promoter regions of *Dlst* and *Idh1*, encoding for cytosolic IDH, further supporting a link between mitochondrial oxidative metabolism, IDH2-mediated αKG synthesis and naïve pluripotency. Transition to a predominantly glycolytic metabolism during EpiLC differentiation in turn limits IDH2-mediated conversion of mitochondrial citrate to αKG, leading to a gradual decrease in intracellular αKG levels. Correspondingly, we ascribe a moderate pluripotency-promoting effect of citrate to IDH2-activity in ESCs, likely resulting in αKG synthesis at the onset of EpiLC induction, before *Idh2* is downregulated. Reduction or depletion of intracellular αKG during EpiLC differentiation conceivably curbs the activity of demethylating enzymes with key roles in preserving a naïve epigenetic state, such as the H3K9me2 demethylases KDM3A and KDM3B (Ko *et al*, 2006; Loh *et al*, 2007) and the TET family enzymes TET1 and TET2 (Costa *et al*, 2013; Hackett & Surani, 2014). Interaction with additional αKG-dependent dioxygenases might further contribute to the naïve pluripotency-promoting effect of αKG.

Collectively, we propose that release from an oxidative metabolic state and diminution of αKG levels are a pre-requisite for the exit from naïve pluripotency and its unique epigenetic state, and successively the acquisition of developmental competence for the germ cell fate. Importantly, we uncouple cell cycle-dependent from cell cycle-independent effects of αKG. To our knowledge, this study is the first to show that limiting cellular proliferation rates during EpiLC induction sustains an ESC-like state. Critically, we demonstrate that αKG can largely replace 2i inhibitors (Ying *et al*, 2008) in maintaining a naïve pluripotent state, suggesting that culture in 2i-conditions may stimulate intracellular αKG production and accumulation. Carey *et al* (2015) recently proposed metabolic re-wiring in 2i culture conditions as a potential mechanism for enabling

glutamine-independent growth of naïve ESCs. However, the molecular basis underlying the re-routing of metabolic flux to increase intracellular αKG levels upon 2i culture remains to be explored. The precise regulatory mechanisms linking 2i culture conditions to mitochondrial respiration merit further investigation into the potential crosstalk between signalling pathways and metabolic state.

While mitochondrial oxidative metabolism declines during the ESC-to-EpiLC transition, super-resolution imaging reveals that, as in EpiSCs (Zhou *et al*, 2012), mitochondria are more elongated and hence likely more developed in EpiLCs than in ESCs, most probably to meet the metabolic requirements of enhanced oxidative metabolism during later stages of development (Folmes *et al*, 2012). Consistently, we find that PGCLCs express higher levels of *Cox7a1* and *Idh2* transcripts, suggesting a boost in mitochondrial oxidative metabolism. Accordingly, activation of mitochondrial oxidative metabolism by 2-DG supplementation results in enhanced PGCLC induction (Hayashi *et al*, 2017). The molecular mechanisms underlying the promotion of PGC fate through stimulation of oxidative metabolism, however, remain to be discovered. Here, we show that αKG largely preserves the histone methylation state underlying the developmental competence for the PGC fate, and extend the interrelation between mitochondrial oxidative metabolism, αKG and epigenetic control from the naïve pluripotent state to PGC development.

Notably, the cellular response to αKG changes during the developmental transition from naïve pluripotency to PGC competency; within the first 24 h of EpiLC differentiation, αKG retains cells in a *Rex1*-high pluripotent state, with low competency for the PGC fate. By contrast, addition of αKG once EpiLCs have acquired PGCLC competency significantly extends the narrow time window of developmental competence for the PGC fate, without affecting the efficiency of the PGCLC induction.

We propose an appropriate balance between H3K9me2 acquisition and H3K27me3 depletion as being a key to the developmental competence for the PGC fate, which is sustained by dm-αKG. Of note, dm-αKG supplementation at the time of competence does not restore the very low H3K9me2 levels as found in naïve ESCs. Instead, through activating αKG-dependent H3K9me2 demethylases, αKG opposes the differentiation-induced H3K9me2 accumulation on certain loci and consequently prevents the genome-wide reduction of H3K27me3 levels. Low levels of DNA demethylation induced by αKG might further promote the spreading of H3K27me3 at high CpG regions (Zylicz *et al*, 2015).

In summary, we suggest that αKG prolongs fleeting developmental states, such as naïve pluripotency and the transient potential for PGC fate, respectively, through safeguarding their particular epigenetic states. It is conceivable that αKG also stabilizes transitory cellular states in other contexts and might potentially provide a universal tool for capturing and expanding short-lived cell states *in vitro* through metabolic modulation.

# Materials and Methods

### Cell lines

C57BL/6 wild-type mouse embryonic stem cells (ESCs; clone C8 was used in this study; Grabole *et al*, 2013) were derived in 2i

conditions as described previously (Nichols *et al*, 2009). For *Prdm1*-GFP ESCs (clone BG5 was used in this study), morula-stage embryos were harvested from uteri of female mice (129 strain) crossed with *Prdm1*-GFP transgene male mice (Ohinata *et al*, 2005). Following 24 h culture in KSOM (Merck) and removal of zona pellucida, blastocyst-stage embryos were harvested on mouse embryonic fibroblasts and cultured in 2i/Lif conditions in GMEM with 10% foetal calf serum (FCS; Gibco). *Rex1*-GFPd2 ESCs were a gift from Tuzer Kalkan (Wray *et al*, 2011; Kalkan *et al*, 2017). *Tet1/2* wild-type and double-knockout (DKO) ESCs (wild-type clone 4 and DKO clone 51) were received from Rudolf Jaenisch (Dawlaty *et al*, 2013). Bill Skarnes and Peri Tate provided E14tg2a wild-type ESCs.

## Cell culture and differentiation

Mouse ESCs were maintained in N2B27, supplemented with 1 μM PD0325901 (Miltenyi Biotec), 3 μM CHIR99021 (Miltenyi Biotec) and 10 ng ml$^{-1}$ LIF (Stem Cell Institute, University of Cambridge (SCI); "2i/Lif" culture conditions; Ying *et al*, 2008) on 0.1% gelatine-coated Nunc cell culture dishes (Thermo Fisher Scientific). For maintaining *Prdm1*-GFP ESCs, foetal calf serum (FCS; Gibco) was added to a final concentration of 5% to 2i/Lif culture medium. Cells were passaged every 2–3 days using TrypLE Express or Accutase (for *Rex1*-GFPd2 ESCs), with media exchange on alternate days. ESCs were grown for at least one passage on dishes coated with 16.67 μg ml$^{-1}$ human plasma fibronectin (FC010; Millipore) in 2i/Lif with 1% knockout serum replacement (KSR, Thermo Fisher Scientific; "2i/Lif/K") before inducing epiblast-like cells (EpiLCs). For differentiation experiments, approximately 25,000 cells per cm$^2$ were plated in fibronectin-coated dishes in EpiLC-inducing culture conditions (N2B27 supplemented with 20 ng ml$^{-1}$ activin A (SCI), 12 ng ml$^{-1}$ bFGF (SCI) and 1% KSR), with daily media change (Hayashi *et al*, 2011; Hayashi & Saitou, 2013). Cells were harvested at $t = 48 \pm 5$ h for downstream assays. For EpiLC differentiation experiments exceeding $t = 48 \pm 5$ h, the initial plating density was adjusted accordingly. For primordial germ cell-like cell (PGCLC) specification, 48 h EpiLCs were aggregated as embryoid bodies in Corning Costar ultra-low attachment 96-well plates (Sigma) at 2,000 cells in 100 μl droplets per well in GMEM BHK-21 (Gibco) with 15% KSR, 0.1 mM NEAA (Thermo Fisher Scientific), 1 mM sodium-pyruvate (Sigma), 2 mM L-glutamine (Sigma), 0.1 mM 2-mercaptoethanol (Thermo Fisher Scientific), 100 U ml$^{-1}$ penicillin and 0.1 mg ml$^{-1}$ streptomycin (Sigma), supplemented with 500 ng ml$^{-1}$ BMP4 (R&D Systems), 500 ng ml$^{-1}$ BMP8a (R&D Systems), 100 ng ml$^{-1}$ SCF (R&D Systems), 50 ng ml$^{-1}$ EGF (R&D Systems) and 10 ng ml$^{-1}$ LIF (SCF; Hayashi *et al*, 2011; Hayashi & Saitou, 2013).

## Single-cell transcriptome profiling

For highly parallel processing of single cells from differentiation time points $t = 0$ (ESCs 2i/Lif/K), $t = 24$ and $t = 48$ h, EpiLCs were induced staggered from C57BL/6 wild-type ESCs (clone C8). Cells were harvested by trypsinization and stained with 2 μg ml$^{-1}$ Hoechst 33342 (Invitrogen, 917368; ESCs 2i/Lif/K), 2.5 μg ml$^{-1}$ CellMask Deep Red plasma membrane stain (Molecular Probes, Life Technologies, C10046; 24 h EpiLCs), and 4 μM ethidium

homodimer-1 and 2 μM calcein (LIVE/DEAD Viability/Cytotoxicity Kit for mammalian cells, Molecular Probes, Life Technologies, L3224; 48 h EpiLCs), respectively, for 20 min at 37°C in 5% CO$_2$ in 1 ml N2B27 with 1% KSR (N2B27/K) each. Labelled cells were washed twice in 500 μl N2B27/K, before combining cells from all three time points in equal numbers for single-cell capture and simultaneous processing using the C1 Single-Cell AutoPrep System (C1 Integrated Fluidic Circuits for mRNA-seq (10–17 μm), Fluidigm, 100-5760; C1 Single-Cell AutoPrep Reagent Kit for mRNA-seq, Fluidigm, 100-6201). Cell identities of single captured cells were deconvoluted based on fluorescent dye labels, using an inverted Olympus fluorescence microscope, before single-cell cDNAs were generated on-chip by SMARTer technology (SMARTer Ultra Low RNA Kit for Illumina Sequencing, Clontech, 634936; Advantage 2 PCR Kit, Clontech, 639206; Ramskold *et al*, 2012). Multiplexed cDNA libraries of single cells were prepared using the Nextera XT DNA Sample Preparation Kit (Illumina, FC-131-1096 and FC-131-1002) and sequenced on the Illumina HiSeq 2000 platform. Extensive quality control analysis was performed, and only 67 single cells that met the following criteria were included for further analysis:
 (i) Cells that could be uniquely identified via fluorescence microscopy and
 (ii) Cells with equal or greater than 6 million uniquely mapping reads.

## Mapping of sequencing reads

Fastq files were filtered for low-quality reads (< Q20), and low-quality bases were trimmed from read ends (< Q20) using the FASTX-Toolkit. Adaptors were removed using CutAdapt (Martin, 2011). The resulting filtered files were mapped to the mouse genome (UCSC mm9) using TopHat 2.0.6 (Trapnell *et al*, 2009; Kim *et al*, 2013) with the UCSC mm9 junction file. BAM files generated from multiple sequencing runs were merged with samtools 0.1.18 (Li *et al*, 2009). Transcript counts and RPKMs were calculated using custom R scripts based on the GenomicRanges Bioconductor library and annotation from the UCSC mm9 junction file. Scripts are available upon request.
FASTX-Toolkit: http://hannonlab.cshl.edu/fastx_toolkit/index.html
CutAdapt: http://journal.embnet.org/index.php/embnetjournal/article/view/200
TopHat: http://www.ncbi.nlm.nih.gov/pubmed/19289445
Samtools: https://www.ncbi.nlm.nih.gov/pubmed/19505943

## Derivation of pseudotime trajectories

Single-cell transcript counts from time points $t = 0$, $t = 24$ and $t = 48$ h during the ESC-to-EpiLC transition were combined, transcripts with no variation removed, and data transformed by $\log_{10}(\text{count} + 1)$. Forty-eight hours EpiLCs with high *Tfcp2l1* expression ($\log_{10}(\text{count} + 1) > 1.5$) were excluded. This left us with 56 out of 67 cells. The R method DESeq2::estimateSizeFactorsForMatrix was used for normalization. To fit the pseudotime model, genes encoding for 135 transcripts, including central regulators of pluripotency, genes associated with epiblast development, epigenetic regulators, transcripts encoding for enzymes within key metabolic pathways and those with the highest ratio of variance between capture time to variance within capture time were chosen (Appendix Table S2). The

DeLorean pseudotime method was applied, using the following hyperparameters: $\sigma_\tau = 8$ h, l = 48 h. The DeLorean model was fit with the No-U-Turn-Sample (NUTS). The null hypothesis (cells were ordered no better than randomly) was rejected by The DeLorean permutation roughness test with $P < 10^{-15}$.

### Quantification of single-cell transcript level changes

For a comprehensive quantification of expression level changes of 478 transcripts encoding for metabolic regulators and control genes, a representative pseudotime for the start and end, respectively, of the ESC-to-EpiLC differentiation was estimated as the median pseudotime for naïve ESCs in 2i/Lif culture conditions and EpiLCs captured at 48 h, respectively: a Gaussian process pseudotime trajectory was fit to each transcript using the cells' pseudotimes inferred from fitting the DeLorean model (see "Derivation of pseudotime trajectories" above). The Kullback–Leibler (KL) divergence between the posterior distributions of the expression trajectory at the representative naïve ESCs and 48 h EpiLCs pseudotimes was calculated as a quantitative measure of change in gene expression. The KL divergence has several properties that make it suitable for this purpose: it is invariant to shifting and scaling of the data; however, it is sensitive to changes in the variance of the pseudotime trajectory. Transcripts were ranked by their KL divergences (Appendix Table S1); higher divergences indicate genes whose distribution of expression has changed the most between the onset and the end point of the ESC-to-EpiLC differentiation.

### Metabolic modulation

For metabolic modulation, 1–10 mM 2-deoxy-D-glucose (2-DG, Sigma-Aldrich, D6134) in $dH_2O$, 5–20 mM sodium dichloroacetate (DCA, Santa Cruz Biotechnology, Inc., sc-203275) in $dH_2O$, 1–4 mM dimethyl alpha-ketoglutarate (dm-αKG, Sigma-Aldrich, 349631), 4 mM sodium citrate dehydrate (Na-citrate, Sigma-Aldrich, W302600) in $dH_2O$ and 4 mM dimethyl succinate (dm-succinate, Sigma-Aldrich, W239607), respectively, were added to cell culture media at the time of plating, with daily media change. For pharmacological modulation during PGCLC differentiation, 4 mM dm-αKG and 500 nM small molecule inhibitor of bone morphogenetic protein (BMP) type I receptors ALK2 and ALK3, LDN-193189 ("iBMP", Stemgent, 04-0074) in DMSO, respectively, were added once at the time of embryoid body aggregation.

### Colony formation assays

Following 48 h culture in EpiLC-inducing conditions in the presence of metabolic modulators (2-DG, DCA, dm-αKG, Na-citrate, dm-succinate), 2,000 cells were plated in fibronectin-coated 6-well plates in 2i/Lif medium with 3% FCS. The next day, cells were rinsed once with 1×PBS and replenished with fresh culture medium. On day 6, cells were fixed with 4% formaldehyde (Thermo Fisher Scientific, PN28906) for 15 min at room temperature and stained for alkaline phosphatase (AP) using Leukocyte Alkaline Phosphatase Kit (Sigma-Aldrich, 86R) according to manufacturer's instructions. AP-positive colonies were quantified and imaged on an upright Zeiss microscope (Stemi SV11), using Leica Application Suite software (v4.1).

### Quantifying cellular proliferation rates

To assess cellular proliferation, ESCs were stained with CellTrace Violet Cell Proliferation Kit (Molecular Probes, Life Technologies, C34557) following manufacturer's instructions for labelling of adherent cells, before EpiLC induction and subsequent quantification of remaining dye levels by flow cytometry. For benchmarking of proliferation rates, dye dilution in the presence of increasing doses (0.1 μm–1 μM) of the cell-permeable cyclin-dependent kinase 4 (CDK4) inhibitor 2-Bromo-12,13-dihydro-5H-indolo[2,3-a]pyrrolocarbazole-5,7(6H)-dione ("CDK4i", Calbiochem, 219476) in DMSO was compared to dye dilution following 4 mM dm-αKG treatment. For gene expression analysis, cells were gated based on CellTrace Violet intensities and collected by fluorescence-activated cell sorting (FACS).

### Flow cytometry

For flow cytometry, cells were re-suspended in 1×PBS with 3% FCS. Flow cytometer analysis was performed on a BD FACScan; data were analysed using BD CellQuest software. FACS sorting was performed on a Moflo (for dye dilution experiments) and SONY SH800 cell sorter (for PGCLC experiments). FACS data were evaluated using FlowJo software.

### Quantitative real-time PCR

RNA was extracted using the RNeasy Mini Kit (Qiagen, 74104; for ESCs and EpiLCs) and Arcturus PicoPure RNA Isolation Kit (Applied Biosystems, Thermo Fisher Scientific, 12204-01; for ESCs and PGCLCs), with on-column DNase digestion (Qiagen, 79254). cDNAs were generated using SuperScript III Reverse Transcriptase (Thermo Fisher Scientific, 18080-044), according to manufacturer's instructions. Quantitative real-time PCR was performed on a QuantStudio 6 Flex Real-Time PCR System (Applied Biosystems), with SYBR Green JumpStart *Taq* ReadyMix (Sigma-Aldrich, S4438), in triplicate for each condition. For each independent biological experiment, data were averaged over technical triplicates and analysed using the comparative Ct method (Schmittgen & Livak, 2008), with transcript levels internally normalized to *ActB* expression levels. Primer pairs used were as follows: *ActB*, forward, 5′-CCCTAAGGCCAACCGTG AAA-3′, reverse, 5′-AGCCTGGATGGCTACGTACA-3′; *Esrrb*, forward, 5′-GGCGTTCTTCAAGAGAACCA-3′, reverse, 5′-CTCCGTTTGGTGA TCTCACA-3′; *Klf4*, forward, 5′-GGGGTCTGATACTGGATGGA-3′, reverse, 5′-CCCCCAAGCTCACTGATTTA-3′; *Tfcp2l1*, forward, 5′-AG GTGCTGACCTCCTGAAGA-3′, reverse, 5′-GTTTTGCTCCAGCTCC TGAC-3′; *Dnmt3b*, forward, 5′-GACGTCCGGAAAATCACCAA-3′, reverse, 5′-GATCATTGCATGGCTTCCA-3′; *Fgf5*, forward, 5′-TA CCCGGATGGCAAAGTCAA-3′, reverse, 5′-ATCCCCTGAGACACAGC AAA-3′; *Lin28b*, forward, 5′-CGAGAGGGAAATCCCTTGGATA-3′, reverse, 5′-CCACTGGCTCTCCTTCTTTCA-3′; *Prdm1*, forward, 5′-G AGGATCTGACCCGAATCAA-3′, reverse, 5′-CTCAACACTCTCATGT AAGAGGC-3′; *Prdm14*, forward, 5′-GCCTGAACAAGCACATGAG A-3′, reverse, 5′-TGCACTTGAAGGGCTTCTCT-3′; *Tfap2c*, forward, 5′-CGCGGAAGAGTATGTTGTTG-3′, reverse, 5′-CGATCTTGATGG AGAAGGTCA-3′; *Klf2*, forward, 5′-ACCAAGAGCTCGCACCTAAA-3′, reverse, 5′-GTGGCACTGAAAGGGTCTGT-3′; *Nanog*, forward, 5′-AC CTGAGCTATAAGCAGGTTAAGAC-3′, reverse, 5′-GTGCTGAGCCCT

TCTGAATCAGAC-3′; *T*, forward, 5′-TCCCGAGACCCAGTTCATA
G-3′, reverse, 5′-TTCTTTGGCATCAAGGAAGG-3′; *Gata6*, forward, 5′-AA
CCCATTCATCCCCGACCAC-3′, reverse, 5′-CTCCTCTCCACGAACG
CTTGT-3′; *Sox7*, forward, 5′-AAACGTCTGGCAGTGCAGAAC-3′,
reverse, 5′-CAGCGCCTTCCATGACTTTCC-3′; *Tet1*, forward, 5′-AGA
TGGCTCCAGTTGCTTATCA-3′, reverse, 5′-ACGCCCCTCTTCATTT
CCAA-3′; *Kdm3a*, forward, 5′-ATTCGAGCTGTTTCCCACAC-3′,
reverse, 5′-TTTCTCCAAGACTCCCCATCA-3′; *Kdm3b*, forward, 5′-C
CATGACCCCAGCAACAAAA-3′, reverse, 5′-TGCACCCCTGAAACTA
GCA-3′; *Cox7a1*, forward, 5′-CGAAGAGGGGAGGTGACTC-3′, reverse,
5′-AGCCTGGGAGACCCGTAG-3′; *Cpt1a*, forward, 5′-GACTCCGCT
CGCTCATTC-3′, reverse, 5′-TCTGCCATCTTGAGTGGTGA-3′; *Idh2*,
forward, 5′-GGATGTACAACACCGACGAGT-3′, reverse, 5′-CGGCCA
TTTCTTCTGGATAG-3′; *Pdk1*, forward, 5′-GTTGAAACGTCCCG
TGCT-3′, reverse, 5′-GCGTGATATGGGCAATCC-3′; *Pdk3*, forward,
5′-AAGCAGATCGAGCGCTACTC-3′, reverse, 5′-TTCACATGCATTA
TCCCTTCC-3′; *Gapdh*, forward, 5′-CCCCAACACTGAGCATCTCC-3′,
reverse, 5′-ATTATGGGGGTCTGGGATGG-3′.

## Small-interfering RNA-mediated knockdown

For combinatorial knockdown via small-interfering RNAs (siRNAs),
275,000 ESCs cultured in 2i/Lif were reverse-transfected with
12.5 nM each of ON-TARGETplus Kdm3a siRNA (GE Healthcare
Lifesciences, L-056510-00-0005) and ON-TARGETplus Kdm3b
siRNA (GE Healthcare Lifesciences, L-065381-00-0005), and
25 nM ON-TARGETplus Non-targeting Pool (GE Healthcare Life-
sciences, D-001810-10-05) as control, respectively, using Dharma-
FECT 1 Transfection Reagent (Dharmacon, T-2001-02) according
to manufacturer's instructions. For each condition, all cells were
plated in one 6-well plate coated with fibronectin in 2i/Lif/K
medium. The following day, cells were replenished with fresh 2i/
Lif/K medium and induced into EpiLCs 1 day later. Knockdown
efficiencies were derived by normalizing *Kdm3a* and *Kdm3b*,
respectively, expression levels in ESCs in 2i/Lif culture conditions
at $t = 48$ h after siRNA transfection to levels prior to siRNA trans-
fection ($t = 0$ h).

## Mitochondria labelling

For staining mitochondria, cells grown on ethanol-cleaned, fibro-
nectin-coated microscope cover glasses (Marienfeld, 0107052) were
washed three times with 1×PBS warmed to 37°C, before 15-min fixa-
tion in 37°C pre-warmed 3% formaldehyde (Thermo Fisher Scien-
tific, PN28906) and 0.1% aqueous glutaraldehyde (Thermo Fisher
Scientific, 50-262-10) in 1×PBS at room temperature, followed by
three rinses in 1×PBS. Cells were permeabilized and blocked for 1 h
at room temperature in 3% BSA and 0.2% Triton X-100 in 1×PBS,
before incubating overnight with primary antibody (rabbit anti-
TOM20, FL-145, Santa Cruz Biotechnology, sc-11415) at a 1:1,000
dilution in 1% BSA and 0.2% Triton X-100 in 1×PBS in a humid
chamber at 4°C. Cells were rinsed three times in 0.05% Triton X-100
in 1×PBS and incubated with secondary antibody (anti-rabbit-IgG-
Atto 647N, Sigma-Aldrich, 40839) diluted 1:500 in 1% BSA and
0.2% Triton X-100 in 1×PBS for 1 h at room temperature in a humid
chamber protected from light. Following three washes in 0.05%
Triton X-100 in 1×PBS, antibody-stained cells were fixed for 10 min
in 3% formaldehyde (Thermo Fisher Scientific, PN28906) and 0.1%

aqueous glutaraldehyde (Thermo Fisher Scientific, 50-262-10) in
1×PBS at room temperature. Cell membranes were labelled with
5 μg ml$^{-1}$ wheat germ agglutinin, Alexa Fluor 488 conjugate (WGA-
488, Thermo Fisher Scientific, W11261) in 1×PBS for 10 min at room
temperature and rinsed three times in 1×PBS, before mounting onto
SuperFrost Plus microscope slides (VWR, 631-0108) in ProLong Gold
antifade reagent (Thermo Fisher Scientific, P36930). Slides were
sealed with nail varnish and stored at 4°C prior to imaging.

## Imaging and analysis

Super-resolution imaging was performed on a custom-built STED
microscope featuring three excitation lines, one fixed depletion
wavelength, fast beam scanning and gated detection. The custom
STED microscope follows closely to the microscope described in
Bottanelli *et al* (2016) (hardware is identical, optical arrangement
differs slightly). All images were acquired with a 100× oil immersion
objective lens (Olympus, UPLSAPO 100XO/PSF). Either a
30 × 30 μm field of view with an image format of 2,048 × 2,048
(14 nm square pixel size) or a 10 × 10 μm field of view ("zoom-in")
with a 1,024 × 1,024 image format (9.8 nm square pixel size) was
used. Unidirectional beam scanning was performed at 16 kHz with
synchronized beam blanking to reduce light exposure. Excitation
laser intensity was approximately 10–20 μW at the microscope side-
port and STED depletion power was 110–120 mW at the microscope
side-port. TOM-20 (Atto 647N) and membrane WGA (Alexa-488)
were imaged simultaneously although the STED depletion beam
only acts on the Atto 647N. Thus, one super-resolved STED mito-
chondria image and one confocal membrane image membrane were
acquired simultaneously. For each line of an image, each line was
scanned either 600 times (10 × 10 μm case) or 650 times
(30 × 30 μm case). For 10 × 10 μm images, acquisition time was
38 s. For the larger 30 × 30 μm images, acquisition time was 83 s.
   To aid visualization, intensity scales (in units of counts) were
adjusted using Fiji software as follows:

ESC2 2i/Lif/K:
TOM-20 (mitochondria, magenta): 0–5 (30 × 30 μm). 0–9
(10 × 10 μm).
WGA-488 (membrane, green): 0–86 (30 × 30 μm).
48 h EpiLCs + DMSO:
TOM-20 (mitochondria, magenta): 0–4
(30 × 30 μm). 0–8 (10 × 10 μm).
WGA-488 (membrane, green): 0–10 (30 × 30 μm).
48 h EpiLCs + dm-αKG:
TOM-20 (mitochondria, magenta): 0–5 (30 × 30 μm). 0–9
(10 × 10 μm).
WGA-488 (membrane, green): 2–48 (30 × 30 μm).

   Bright-field and epifluorescence images of cells were acquired on
an inverted Olympus microscope with Leica Application Suite soft-
ware (v4.1) and processed using Fiji software. Fluorescent image
intensity scales (in units of counts) were adjusted equally.

## Western blot analysis

Cells were harvested, re-suspended in 50 mM Tris–HCl (pH 8.0)
supplemented with 1% SDS, 10 mM EDTA, 1× protease inhibitor

cocktail (Roche) and lysed by 10 min of incubation on ice. Cell lysates were cleared through 15 min of centrifugation at 13,000 *g*, protein concentrations (determined using the Bicinchoninic Acid Kit, Sigma-Aldrich) were adjusted, and samples were incubated for 5 min at 95°C following addition of Laemmli buffer. Proteins were separated on 12% polyacrylamide gels using the Mini-PROTEAN system (Bio-Rad) and transferred to an Immobilon-P transfer membrane (Millipore). Following 2 h of blocking in 5% skimmed milk, the membranes were incubated with primary antibodies, diluted in 5% BSA, 0.01% TBST overnight at 4°C. Primary antibodies used in this study were as follows: rabbit anti-H3K27me3 (Cell Signaling Technology, C36B11; 1:5,000), mouse anti-H3K9me2 (Abcam, ab1220; 1:5,000), rabbit anti-H3 (Abcam, ab1791; 1:10,000), goat anti-DNMT3b (Santa Cruz Biotechnology, sc-10235; 1:1,000) and rabbit anti-IDH2 [Abcam, ab129180 (EPR7576); 1:1,000].

Histone antibody binding was visualized using IRDye 680RD, goat anti-mouse IgG IRDye 680RD, goat anti-rabbit IgG IRDye 800CW and goat anti-mouse IgG IRDye 800CW, respectively, secondary antibodies (LI-COR; 1:2,000 in 5% skimmed milk, 0.01% TBST) and the LI-COR Odyssey CLx system. DNMT3b and IDH2, respectively, antibody binding was detected by horseradish peroxidase-conjugated anti-goat IgG (Dako; 1:2,000 in 5% skimmed milk, 0.01% TBST) and anti-rabbit IgG (Dako; 1:5,000 in 5% skimmed milk, 0.01% TBST), respectively, in conjunction with the Western Detection System (GE Healthcare).

### ChIP-qPCR analysis

Native ChIP (nChIP) was performed as previously described (Brind'Amour *et al*, 2015). Briefly, cells were washed, dissociated and stored in nuclear storage buffer (Nuclei Isolation Kit; Sigma-Aldrich) at −80°C prior to lysis in digestion buffer [1× MNase buffer (NEB); 2.5 mM DTT, 6.25% PEG-6000, 2.5 U MNase (NEB)]. Following chromatin pre-clearance through protein A/G Dynabeads (Thermo Fisher Scientific), the antibody–bead complex was formed by 2-h incubation with antibody in IP buffer (20 mM Tris–HCl pH 8.0; 2 mM EDTA; 150 mM NaCl; 0.1% Triton X-100), with protein inhibitor cocktail (Roche) at 4°C. Chromatin and antibody-bead complex were then inculcated overnight at 4°C and washed, and purified DNA was quantified by qPCR on a QuantStudio 6 Flex Real-Time PCR System (Applied Biosystems). Antibodies used for nChIP experiments were as follows: anti-H3K27me3 (Cell Signaling Technology; C36B11; lot 8); anti-H3K9me2 (Abcam; ab1220; lot GR212253-7); rabbit IgG (Santa Cruz Biotechnology; sc-2027; lot H2615); mouse IgG (Santa Cruz Biotechnology; sc-2025; lot G2314). H3K9me2 and H3K27me3, respectively, occupancy was investigated in putative enhancer regions of candidate genes, based on three published datasets (Ma *et al*, 2011; Buecker *et al*, 2014; Zylicz *et al*, 2015). Primer pairs used were as follows: *Esrrb* enhancer, forward, 5′-AGGT TTGAATGGGACAGGAG-3′, reverse, 5′-GATTGCACATCAAGGACT GG-3′; *Arid5b* enhancer, forward, 5′-GGATTCAGAGAGCAAGCACA-3′, reverse, 5′-TGCTTCTGCAGGAATCTCAG-3′; *Tfap2c* enhancer, forward, 5′-GCGCTTAGGTCGCTTGGATA-3′, reverse, 5′-CTCGAACA CTTGGAGTCGGG-3′; *Nanog* enhancer, forward, 5′-TTCAGTCAGG CTGGGCAATG-3′, reverse, 5′-CCTCAACTGCTGCCACACTA-3′; *Prdm14* enhancer, forward, 5′-AAGCAGCAGGGTGGAGATAA-3′, reverse, 5′-AAATGGGCTGCTAAGTGCAT-3′; *Pdgfa* enhancer,

forward, 5′-CCTCATCTTCCTCCTTCCAC-3′, reverse, 5′-AAATCAGAC AGGCAGGGTGT-3′; *Tfap2c* promoter, forward, 5′-CAGCCAGATA CAGCTTCGGG-3′, reverse, 5′-GATTCCGAGAAGGAGTCCGC-3′.

### OSN binding site analysis

For transcription factor occupancy analysis, binding of OCT4, SOX2 and NANOG (OSN) within a region of 20 kb upstream to 4 kb downstream of the transcriptional start sites of genes of interest was investigated, based on ChIP-seq summary data from three separate publications (Chen *et al*, 2008; Marson *et al*, 2008; Whyte *et al*, 2013). A gene was defined as occupied by OSN if all three factors bound the region in the summary data from at least two of the three publications.

### Quantification and statistical analysis

Microsoft Excel was used for statistical evaluation of gene expression (qRT–PCR), flow cytometer analysis and colony formation assays. Data were analysed using unpaired (heteroscedastic) 1-tailed Student's *t*-tests. For comparing fold changes in gene expression levels, statistical analysis was performed on $\log_{10}$-transformed data. Statistical details of the experiments, such as the number of independent biological replicates, definition of centre, dispersion and significance are reported in the figure legends. Data are represented as mean ± 1 SEM. Significance levels are denoted as follows: *$P \leq 0.05$; **$P \leq 0.01$; ***$P \leq 0.005$. *P*-values for all statistically evaluated experiments are listed in Appendix Table S3.

## Data availability

The single-cell RNA-seq data reported here have been deposited in GEO under accession number GSE107761.

**Expanded View** for this article is available online.

### Acknowledgements

We thank M. Lynch for help with single-cell processing on the Fluidigm platform; S. Leigh-Brown for generating single-cell RNA-seq libraries; J. Hadfield and D. Bentley for single-cell RNA-sequencing; N. Miller for EpiLC FACS sorting; C. Lee for preparing N2B27 stem cell media; G. Sirinakis for help with super-resolution imaging; T. Kalkan, T. Kobayashi, P. Tate and W. Skarnes for ESC lines; C. Bradshaw, G. Allen and S. Dietmann for mapping sequencing reads; B. Goettgens and S. Teichmann for discussion on single-cell expression analysis; M. Mueschen for discussion on cellular metabolism; U. Gunesdogan, C. Penfold, J. van den Ameele and S.J. Maerkl for critically reading the manuscript and for discussion; and T. Bollenbach and all members of the Surani laboratory for input. JT was supported by the Austrian Academy of Sciences, the Wellcome Trust and the Swiss National Fund for Science; WHG by EMBO and the Wellcome Trust; JR and LW by the UK Medical Research Council; and EA and BDS by the Wellcome Trust. MAS is a Wellcome Senior Investigator. Work at the Gurdon Institute is supported by a core grant from The Wellcome Trust and Cancer Research UK.

### Author contributions

JT designed the experiments; performed cell culture, single-cell processing for RNA-seq, flow cytometry, qRT–PCR, immunofluorescence, imaging, data

analysis and graphical representations; and wrote the paper. WHG performed Western blot and ChIP-qPCR analyses, qRT–PCR, data analysis and graphical representations of the experimental results. JR and LW performed pseudotime, statistical and binding site analyses. EA built the STED microscope and helped with super-resolution imaging. FB, CM and FT performed the GPLVM analysis. BDS provided experimental support. MAS supervised the study.

## Conflict of interest

The authors declare that they have no conflict of interest.

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
