## [Review Process File · The EMBO Journal]

Metabolic regulation of pluripotency and germ cell fate through α -ketoglutarate

Julia Tischler, Wolfram H. Gruhn, John Reid, Edward Allgeyer, Florian Buettner, Carsten Marr, Fabian Theis, Ben D. Simons, Lorenz Wernisch, M Azim Surani

Review timeline:

Submission date:	27th Mar 2018
Editorial Decision:	25th Apr 2018
Revision received:	31st Jul 2018
Editorial Decision:	17th Aug 2018
Revision received:	24th Aug 2018
Accepted:	27th Aug 2018

Editor: Daniel Klimmeck

Transaction Report:

1st Editorial Decision

25th Apr 2018

Thank you for the submission of your manuscript (EMBOJ-2018-99518) to The EMBO Journal. Please accept my apologies for the delay in getting back to you due to delayed referee feedback. Your study has been sent to three referees, and we have received reports from all of them, which I enclose below.

As you will see, the referees acknowledge the potential high interest and novelty of your work, although they also express a number of concerns that will have to be addressed before they can support publication of your manuscript in The EMBO Journal. While referee #3 is overall positive and supportive, referee #1 states concerns about the claims made on relevance of aKG and oxidative phosphorylation for stem cell fate determination and asks you to corroborate your analyses (referee #1, pts 1,3). Along the same lines, this referee also requests analysis of protein levels and activities of enzymes involved in aKG synthesis (referee #1, pt. 2). Referee #2 agrees in that the causal link between aKG and control of differentiation is not sufficiently supported by the current data, and points to subtlety of effects and inconsistencies in the data. In addition, all referees list a number of technical and methods issues on assays used and controls made, as well as incomplete documentation, points that need to be addressed to achieve the level of robustness required for The EMBO Journal.

I judge the comments of the referees to be generally reasonable and we are in principle happy to invite you to revise your manuscript experimentally to address the referees' comments. I agree that your resource would strongly benefit from more in-depth analysis of the functional implication of aKG as opposed to other differential regulators.

REFeree COMMENTS:

Referee #1:

In this manuscript 'Metabolic regulation of pluripotency and germ cell fate through alpha-ketoglutarate', Tischler et al investigate the transcriptomic dynamics during stem cell differentiation and influences of alpha-ketoglutarate (aKG) on stem cell fate determination. This is generally a solid study conducted at a high standard. The findings on aKG's roles in different stages of stem cell differentiation are interesting. However, the evidence that distinguish aKG from other factors that also change during stem cell differentiation are less solid than the other parts of this study.

1. As I mentioned above, the evidence from single cell RNA-seq are not strong enough to support the importance of alpha-ketoglutarate in stem cell fate determination. The only direct evidence is the down-regulation of *Idh2* and up-regulation of *Dlst* during the ESC-to-EpiLC differentiation, but there are also other metabolic genes that have changes in their expression levels in this process. Moreover, the correlation between *Dlst* expression and the pseudotime (Fig 3A) is very subtle, suggesting that *Dlst* is not likely to be among the metabolic genes with the most dramatic changes. A comprehensive comparison among all genes considered in the pseudotime analysis will be helpful to clarify the importance of aKG.

2. The authors conclude that evidence demonstrating changes in expression levels of genes coding for enzymes involved in aKG generation distinguishes their study from previous recent reports showing the importance of aKG in regulating pluripotency (Fig EV1D). However, gene expression is not a direct indicator of enzyme level or activity. Thus, it would be more informative if protein level or enzyme activity was quantified and presented in these contexts.

3. Inhibition of glycolysis by 2-DG is used to validate the importance of OXPHOS in maintaining naïve pluripotency. This evidence is indirect and there is no other evidence showing that the inhibition of glycolysis results in enhanced OXPHOS.

4. More details of the computational methods are necessary. For instance, the software used to obtain the transcript counts needs to be reported. For the pseudotime analysis, it is also not clear what criterion is used to identify genes that change during the differentiation.

5. The authors claim that aKG can safeguard the transient epigenomic status during differentiation by maintaining histone methylation levels (Fig 4E). aKG is well known as a substrate of the aKG dependent histone and DNA demethylases. Thus, by intuition, adding aKG should lead to enhanced activity of these enzymes and loss of the histone methylation marks. However, Fig 4E shows that addition of aKG results in increased H3K27me3 levels. This is interesting but some discussion on possible mechanisms will be helpful.

6. The color scheme used in some figures (Fig 2B, 3B, etc) needs to be changed. It is very hard to see the difference between the two green colors used in these figures.

Referee #2:

Developmental potential of pluripotent stem cells (PSCs) is regulated by sophisticated molecular networks controlling their gene expression, epigenetic status as well as metabolic status. In this study, the author examined roles of energy metabolic status during development of naïve pluripotential ES cells (ESCs) into PGCLCs via primed epiblast-like cells (EpiLCs) in culture. They first found that glycolysis- and oxidative phosphorylation-related enzyme genes were up- and down-regulated, respectively, in the course of ESC differentiation to EpiSCs by single-cell RNA-seq, and also showed inhibition of glycolysis resulted in retention of ESCs undergoing conversion to EpiLCs in naïve status. The results together indicate the importance of glycolysis in primed PSCs, which is consistent to previous studies.

The authors then focused on aKG, a metabolite in the TCA cycle, because of down- and up-regulation of an aKG-producing enzyme gene, *Idh2* and aKG-catabolizing enzyme gene, *Dlst*, respectively, during EpiLC development from ESCs. Further investigation concerning roles of aKG indicated that aKG repressed conversion of ESCs to EpiLCs and partially replaced 2i for maintenance of naïve pluripotency of ESCs. The results also showed that addition of Citrate, an upstream metabolite to aKG, but not a down-stream metabolite Succinate, repressed EpiLC development from ESCs, indicating the importance of levels of aKG among the TCA cycle metabolites for regulating EpiLC development. The authors then demonstrated that repression of EpiLC development by aKG did not necessarily depend on repression of cell cycle progression. aKG is a co-factor of H3K9me2 demethylases KDM3A and KDM3B and TET enzymes involved in DNA demethylation, and the results demonstrated that knock-down of *Kdm3a/b* or *Tet1/2* repressed

the inhibitory effect of aKG for EpiLC development from ESCs, suggesting that aKG functions on EpiLC development via H3K9me2 and DNA methylation. The authors finally tested roles of aKG on PGCLC differentiation from EpiLCs, and found that aKG stimulated PGCLC formation from EpiLCs and partially replaced BMP4. In addition, aKG sustained the PGCLC competence in EpiLCs probably via affecting histone and DNA methylation.

This study revealed the importance of levels of aKG on EpiLC development from ESCs, PGCLC competence as well as PGCLC differentiation from EpiLCs, which should be highly interesting and important to understand roles of energy metabolisms on EpiLC and PGCLC development. Meanwhile experimental evidence showing direct linkage between aKG and regulation of epigenetic status in EpiLCs is weak.

Specific comments:

1. Fig.4E, Fig. EV6A, page 10, 2nd paragraph; Expected effects of addition of aKG are decreased H3K9me and H3K27me3. H3K9me2 is not significantly changes or is slightly decreased, if any, but H3K27me3 is clearly increased in dm-aKG treated cells compared with control cells both at 48 and 72 hr in Fig. 4E, which is not consistent with the expected functions of aKG. In addition, if aKG results in decreased H3K9me2, it may be more suitable to naïve status than primed EpiLCs status, which may also be inconsistent with prolonged PGCLC competence in EpiLCs in the presence of aKG. Please address those issues.

2. Fig.4, page 8-; Addition of dm-aKG enhanced PGCLC differentiation from EpiLCs, and it suggests that aKG is increased during PGCLC development. In that case, is the expression of *Idh2* and *Dist* up- and down-regulated, respectively, in the course of PGCLC development?

Minor points:

1. Fig. EV4I; The graph shows relative colony formation in *Kdm3a/b*-KD condition over control with or without aKG, but in the absence of dm-KG, colonies may be rarely formed, and it is important to show increased colony formation by aKG without *Kdm3a/b*-KD and decreased aKG-induced colony formation by *Kdm3a/b*-KD. Therefore it may be more suitable to compare ratios of colony number in total number of plated cells in each condition.
2. The color codes may be inverted in the graphs in Fig. 3C, Fig.3H right panel.
3. Fig.3G left panel; EpiLCs in the legend may be ESCs w/o 2i.
4. page 9, 3rd paragraph, lines 1- 2; 'Addition of dm-aKG up to 24h after-----' may be 'Addition of dm-aKG from 24h to 48h after-----'.
5. page 12, lines 2-3; Please cite a reference or show data concerning OSN binding in *Idh2*.

Referee #3:

This is an excellent and novel manuscript that for the first time systematically tests the effect of aKG co-factor on distinct states of pluripotency, during their transition and in PGCs. This is a much needed study and set of results following the initial study by Carey et al. Nature 2015 showing that Glutamine helps pluripotency via aKG production, however since then systematic characterization of aKG on pluripotency was missing (and in different states and during their transitions).

The authors use exacting reporters and cutting edge differentiation protocols (e.g. REX1-dGFP, *Blimp1*-mVenus and PGC reporters) to test the result and responsive genes to aKG. They establish beyond any doubt that aKG supports naive and PGC state and not the primed states. They do not stop there, but also show the importance of TET enzymes and *KDm* members to directly dictate the epigenetic state. They also establish difference in TCA cycle gene expression during naive to primed state, that supports the rationale for having aKG levels high in naive state.

The manuscript is very well written, and all conclusions are strongly supported by the results. I have no meaningful comments to add or experiments to requests, and support publishing this exciting and thorough study.

Rebuttal

General comments

We are most grateful for the Editorial guidance for the revisions, and thank the Reviewers for their suggestions and constructive comments. In response, we have performed additional experiments and made appropriate changes to the manuscript. These include comprehensive statistical analysis on our single-cell expression data to quantify transcript level changes during the conversion to EpiLC fate with respect to the 478 regulators with roles in cellular metabolism. Importantly, we measured levels of the α KG-generating enzyme IDH2 during the ESC-to-EpiLC transition to strengthen the connection between IDH2-mediated α KG synthesis and the regulation of pluripotency. We further confirm elevated IDH2 protein levels as readout of TCA cycle activity following glycolytic inhibition through 2-DG. We provide additional support for a link between α KG and the epigenetic state of EpiLCs through the locus-specific analysis of H3K9me2 and H3K27me3 modifications by ChIP-qPCR. Accordingly, we discuss the increase in H3K27me3 levels following dm- α KG supplementation and clarify the context-dependent roles of α KG in maintaining naïve pluripotency, and prolonging PGCLC competency, respectively.

Our specific comments follow:

Referee #1

In this manuscript 'Metabolic regulation of pluripotency and germ cell fate through alpha-ketoglutarate', Tischler et al investigate the transcriptomic dynamics during stem cell differentiation and influences of alpha-ketoglutarate (α KG) on stem cell fate determination. This is generally a solid study conducted at a high standard. The findings on α KG's roles in different stages of stem cell differentiation are interesting. However, the evidence that distinguish α KG from other factors that also change during stem cell differentiation are less solid than the other parts of this study.

1. As I mentioned above, the evidence from single cell RNA-seq are not strong enough to support the importance of alpha-ketoglutarate in stem cell fate determination. The only direct evidence is the down-regulation of *Idh2* and up-regulation of *Dl1st* during the ESC-to-EpiLC differentiation, but there are also other metabolic genes that have changes in their expression levels in this process. Moreover, the correlation between *Dl1st* expression and the pseudotime (Fig 3A) is very subtle, suggesting that *Dl1st* is not likely to be among the metabolic genes with the most dramatic changes. A comprehensive comparison among all genes considered in the pseudotime analysis will be helpful to clarify the importance of α KG.

We carried out a comprehensive statistical analysis of changes during the ESC-to-EpiLC transition of 478 transcripts linked to cellular metabolism, including 83 α KG-dependent dioxygenases, and of the key pluripotency (*Esrrb*, *Klf4*, *Nanog*, *Sox2*, and *Tfcp2l1*) and epiblast

fate (*Dnmt3b*, *Fgf5*, *Lin28b*) markers as references. We use the Kullback-Leibler (KL)-divergence between the genes' expression at an early (ESCs in 2i/Lif culture conditions) and a late (48h EpiLCs) stage, respectively, as a metric to quantify gene expression changes. Higher KL-scores represent larger changes in transcript levels. The KL-divergence has several properties that make it suitable for this purpose: it is invariant to shifting and scaling of the data; however, it is sensitive to changes in the variance of the pseudotime trajectory. Ordering all 478 transcripts by their KL-divergences, *Idh2* and *Dlst* regulators have high KL-scores ranked as the 23rd and 58th, respectively, supporting substantial changes in expression during the ESC-to-EpiLC transition (see Appendix Table S1). Importantly, we identify *Idh2* as the transcript with the top KL-score amongst all the TCA cycle genes analyzed, suggesting that the dynamic regulation of intracellular α KG levels through modulation of oxidative mitochondrial metabolism might be important for EpiLC differentiation. Indeed, we confirm this hypothesis through in-depth experimental validation. Genes with key roles in oxidative and glycolytic metabolism, respectively, rank comparably by our metric (i.e. KL-scores for *Cox7a1*, *Cpt1a*, and *Pdk1* rank at 13th, 17th, and 32nd, respectively).

We provide this data set as a rank-ordered list (by descending KL-scores) as Appendix Table S1. Accordingly, we refer to the comprehensive quantitative analysis in the main text (see page 4, first paragraph, lines 10-11, page 5, last paragraph, lines 1-3, and page 12, first paragraph, line 3). We provide details on the statistical analysis in the updated Methods section (see page 18, last paragraph). Furthermore, we acknowledge the less-pronounced change in *Dlst* transcript levels and now refer to it as 'slight' up regulation in the main text (see page 5, last paragraph, lines 2-3).

2. The authors conclude that evidence demonstrating changes in expression levels of genes coding for enzymes involved in α KG generation distinguishes their study from previous recent reports showing the importance of α KG in regulating pluripotency (Fig EV1D). However, gene expression is not a direct indicator of enzyme level or activity. Thus, it would be more informative if protein level or enzyme activity was quantified and presented in these contexts.

We measured IDH2 protein levels in naïve ESCs and at t=24h, 48h, and 72h during the EpiLC differentiation. Western blot analysis confirms the pronounced reduction of the α KG-generating enzyme during the ESC-to-EpiLC transition (see Fig EV3A to complement the pseudotime trajectory for *Idh2* in Fig 3A). We amend the text in our manuscript accordingly (see page 6, lines 2-3, and page 12, second paragraph, lines 3-5).

3. Inhibition of glycolysis by 2-DG is used to validate the importance of OXPHOS in maintaining naïve pluripotency. This evidence is indirect and there is no other evidence showing that the inhibition of glycolysis results in enhanced OXPHOS.

Principally, restriction of glycolysis requires cells to use alternative metabolic pathways to meet their energetic demands. We verify an increased dependency on mitochondrial oxidative metabolism following glycolytic inhibition by 2-DG in the context of the ESC-to-EpiLC transition: we confirm elevated IDH2 protein levels as readout of enhanced TCA cycle activity following 2-DG supplementation during the EpiLC differentiation (see Fig EV2A, referred to on page 5, line 6). A previous report indicated increased mitochondrial respiration after addition of 2-DG to mESCs cultures (Zhou et al., 2012). Note that up regulation of oxidative metabolism as an adaptive response to compensate for glycolytic inhibition by 2-DG has also been observed in various tumor-derived cell lines (Pusapati & Settleman, 2016, Sottnik, Lori et al., 2011).

4. More details of the computational methods are necessary. For instance, the software used in obtain the transcript counts needs to be reported. For the pseudotime analysis, it is also not clear what criterion is used to identify genes that change during the differentiation.

We include details on the software used to obtain single-cell transcript counts in the Revised Methods section (see page 17, first full paragraph, lines 6-8). In addition, we provide details on the computational and statistical methods employed for the comprehensive quantitative pseudotime analysis of 478 transcripts implicated in metabolic regulation (see page 18, last paragraph, and Appendix Table S1; please also see point 1 above).

5. The authors claim that aKG can safeguard the transient epigenomic status during differentiation by maintaining histone methylation levels (Fig 4E). aKG is well known as a substrate of the aKG dependent histone and DNA demethylases. Thus, by intuition, adding aKG should lead to enhanced activity of these enzymes and loss of the histone methylation marks. However, Fig 4E shows that addition of aKG results in increased H3K27me3 levels. This is interesting but some discussion on possible mechanisms will be helpful.

The transition from pre- to post-implantation epiblast *in vivo* entails rapid accumulation of the repressive H3K9me2 mark at promoters, gene bodies, and enhancers of developmentally linked genes, and concomitantly, a genome-wide reduction and redistribution of the repressive

H3K27me3 mark (Kurimoto, Yabuta et al., 2015, Zylicz, Dietmann et al., 2015). The ESC-to-EpiLC differentiation system *in vitro* faithfully recapitulates the histone methylation dynamics observed during this developmental transition (see Fig EV3A).

We propose that dm- α KG supplementation during the EpiLC induction sustains KDM3A and KDM3B activity, thereby counteracting the locus-specific accumulation of H3K9me2 (see Fig EV4H, I and Fig 4F). In addition, α KG likely supports the low-DNA methylation state by enhancing the efficiency of the DNA hydroxylases TET1 and TET2 (see Fig EV4J). Consequently, α KG plausibly prevents the genome-wide reduction in H3K27me3 that normally occurs during epiblast development, by further promoting its spreading at high CpG- regions (Zylicz et al., 2015).

Collectively, we show that α KG results in lower global H3K9me2 levels, which is consistent with a reduction of H3K9me2 levels at certain cis-regulatory elements (see Fig 4F; please also see point 1 below), and conversely, increased H3K27me3 levels (Fig 4E), in line with the characteristic epigenetic state of a relatively naïve cell state (Fig EV3A).

We discuss the possible counter-intuitive increase in H3K27me3 levels following addition of dm- α KG in the manuscript (see page 10, last paragraph, lines 7-10, and page 14, last paragraph).

Please also see our comments in response to Referee #2, point 1 below.

6. The color scheme used in some figures (Fig 2B, 3B, etc) needs to be changed. It is very hard to see the difference between the two green colors used in these figures.

We have amended the color scheme to improve the visibility of the differences in Fig 2B, 3B, and 3G.

Referee #2:

Developmental potential of pluripotential stem cells (PSCs) are regulated by sophisticated molecular networks controlling their gene expression, epigenetic status as well as metabolic status. In this study, the author examined roles of energy metabolic status during development of naïve pluripotential ES cells (ESCs) into PGCLCs via primed epiblast-like cells (EpiLCs) in culture.

They first found that glycolysis- and oxidative phosphorylation-related enzyme genes were up-

and down-regulated, respectively, in the course of ESC differentiation to EpiSCs by single-cell RNA-seq, and also showed inhibition of glycolysis resulted in retention of ESCs undergoing conversion to EpiLCs in naïve status. The results together indicate the importance of glycolysis in primed PSCs, which is consistent to previous studies.

The authors then focused on aKG, a metabolite in the TCA cycle, because of down- and up-regulation of an aKG-producing enzyme gene, *Idh2* and aKG-catabolizing enzyme gene, *Dist*, respectively, during EpiLC development from ESCs. Further investigation concerning roles of aKG indicated that aKG repressed conversion of ESCs to EpiLCs and partially replaced 2i for maintenance of naïve pluripotency of ESCs. The results also showed that addition of Citrate, an up-stream metabolite to aKG, but not a down-stream metabolite Succinate, repressed EpiLC development from ESCs, indicating the importance of levels of aKG among the TCA cycle metabolites for regulating EpiLC development. The authors then demonstrated that repression of EpiLC development by aKG did not necessarily depend on repression of cell cycle progression. aKG is a co-factor of H3K9me2 demethylases KDM3A and KDM3B and TET enzymes involved in DNA demethylation, and the results demonstrated that knock-down of *Kdm3a/b* or *Tet1/2* repressed the inhibitory effect of aKG for EpiLC development from ESCs, suggesting that aKG functions on EpiLC development via H3K9me2 and DNA methylation. The authors finally tested roles of aKG on PGCLC differentiation from EpiLCs, and found that aKG stimulated PGCLC formation from EpiLCs and partially replaced BMP4. In addition, aKG sustained the PGCLC competence in EpiLCs probably via affecting histone and DNA methylation.

This study revealed the importance of levels of aKG on EpiLC development from ESCs and on PGCLC competence as well as PGCLC differentiation from EpiLCs, which should be highly interesting and important to understand roles of energy metabolisms on EpiLC and PGCLC development. Meanwhile experimental evidence showing direct linkage between aKG and regulation of epigenetic status in EpiLCs is weak.

We greatly appreciate this feedback; we address the linkage between \$\alpha\$ KG and epigenetic state in more detail (see below).

Specific comments:

1. Fig.4E, Fig. EV6A, page 10, 2nd paragraph; Expected effects of addition of aKG are decreased H3K9me and H3K27me3. H3K9me2 is not significantly changes or is slightly decreased, if any, but H3K27me3 is clearly increased in dm-aKG treated cells compared with control cells both at 48 and 72 hr in Fig. 4E, which is not consistent with the expected functions of aKG

We acknowledge the comment and refer to our comment above (Referee #1, point 5). The global changes in H3K9me2 levels following dm- α KG treatment are mild (Fig 4E), however, the conversion from naïve to primed pluripotency involves the acquisition of the H3K9me2 mark at promoters, gene bodies, and enhancers of some developmentally linked genes (Zylicz et al., 2015). We address this with reference to the locus-specific effects of dm- α KG supplementation during the EpiLC induction, by analyzing H3K9me2 and H3K27me3 modifications by ChIP-qPCR (see Fig 4F, Fig EV6D).

We find that dm- α KG opposes the accumulation of H3K9me2 on certain enhancer elements associated with the genes for the naïve pluripotent state (Zylicz et al., 2015), such as *Esrrb*, *Arid5b*, *Prdm14*, and *Nanog*. Consistent with this, levels for the repressive H3K27me3 mark are higher at these pluripotency-associated genes, except for *Prdm14*, where we detected no change. By contrast, we observe an inverse pattern, with high H3K9me2 and low H3K27me3, respectively, marking in the cis-regulatory region of *Pdgfa*. Notably, on loci, such as in the enhancer region of the germ cell-associated gene *Tfap2c*, H3K9me2 levels show an increase, while there also is an increase in the repressive H3K27me3 mark. This is consistent with repression of germline genes by both H3K9me2 and H3K27me3 during epiblast development (Kurimoto et al., 2015).

Altogether, while there is reduction in the global H3K9me2 levels following dm- α KG supplementation during the EpiLC induction, we observe a locus-specific effect of dm- α KG. Thus, a subset of cis-regulatory regions, in particular enhancer elements of pluripotency-associated genes have low levels of H3K9me2, but other regulatory regions, such as enhancers of the germ cell-associated gene *Tfap2c*, or the platelet-derived growth factor subunit A (*Pdgfa*), respectively, show an increase in H3K9me2 levels, irrespective of dm- α KG treatment. The locus-specific effect might reflect the selective recruitment of α KG-dependent H3K9me2 demethylases and the subtle changes in global H3K9me2 levels.

We present the results of these new ChIP-qPCR data on representative loci (*Esrrb*, *Arid5b*, *Tfap2c*, *Nanog*, *Prdm14*, and *Pdgfa*) in Fig 4F and Fig EV6D and in the text (see page 11, first full paragraph, page 11, second full paragraph, lines 4-8, and page 14, last paragraph, lines 4-6); we provide technical details in the Methods section (see page 24, last paragraph, continued on page 25).

In addition, if α KG results in decreased H3K9me2, it may be more suitable to naïve status than primed EpiLCs status, which may also be inconsistent with prolonged PGCLC competence in EpiLCs in the presence of α KG. Please address those issues.

We agree that α KG plays an important role in maintaining naïve pluripotency, as we demonstrate through its inhibitory effect on the conversion to the primed pluripotent state and its ability to partially replace 2i inhibitors in sustaining an embryonic stem cell (ESC) state. The addition of dm- α KG even at t=24h of EpiLC stimulation, results in the majority of cells retaining a *Rex1*-GFP positive state (see attached Figure); consequently, there is a reduction in the efficiency of PGCLC specification (Fig EV6A, EV6B). By contrast, the addition of dm- α KG at t=48h of EpiLC differentiation, at the time when cells start to become competent for the PGCLC fate, promotes and extends the duration of the competent state to 72h, but without a change in the efficiency of the PGCLC induction (Fig 4C, Fig EV6C). Note that in the control EpiLCs, the competent state for the specification of *Prdm1*-GFP positive PGCLCs largely declines after 48h.

We propose a relative balance between H3K9me2 and H3K27me3 as being a key to the developmental competence for the PGC fate, which is apparently sustained for a longer period by dm- α KG. Of note, dm- α KG supplementation at the time of competence does not restore the very low H3K9me2 levels to those detected in naïve ESCs. Instead, α KG conceivably counteracts the differentiation-stimulating bFGF & ActivinA signaling and maintains equilibrium between H3K9me2 removal through activating α KG-dependent H3K9me2 demethylases, and differentiation-induced H3K9me2 accumulation at certain loci, and consequently, prevents the global reduction of H3K27me3. This proposal merits further investigation in the future.

We discuss the dual functions of α KG in sustaining the naïve pluripotent state and in extending the developmental competence for the PGC fate, respectively, in detail (see page 14, second and third full paragraphs, and page 15, first full paragraph).

2. Fig.4, page 8-; Addition of dm- α KG enhanced PGCLC differentiation from EpiLCs, and it suggests that α KG is increased during PGCLC development. In that case, is the expression of *Idh2* and *Dist* up- and down-regulated, respectively, in the course of PGCLC development?

We investigated expression levels of key regulators implicated in oxidative metabolism (*Cox7a1*, *Cpt1a*), α KG synthesis (*Idh2*), and glycolysis (*Pdk1*, *Pdk3*) during PGCLC development by qRT PCR analysis. We find *Cox7a1* and *Idh2* to be up regulated in day4 PGCLC, in line with an enhanced oxidative metabolism and increased α KG generation during PGC development. This is

also consistent with a recent report from our group, showing that *Idh2* transcript levels reach a maximum in E9.5 *ex vivo*-isolated PGCs (Hackett, Sengupta et al., 2013).

We include the new data set as Fig EV5A and refer to it in the main text (see page 9, first paragraph, lines 2-8, and page 14, first paragraph, lines 5-6).

Minor points:

1. Fig. EV4I; The graph shows relative colony formation in Kdm3a/b-KD condition over control with or without aKG, but in the absence of dm-KG, colonies may be rarely formed, and it is important to show increased colony formation by aKG without Kdm3a/b-KD and decreased aKG-induced colony formation by Kdm3a/b-KD. Therefore it may be more suitable to compare ratios of colony number in total number of plated cells in each condition.

We present bright-field images of AP-stained colonies at the bottom of the figure panel, which represent the low colony forming ability of DMSO-treated control cells, the enhanced colony formation after addition of dm- α KG during EpiLC induction, and the attenuation of the dm- α KG-induced effect upon knockdown of *Kdm3a/b*, respectively. However, we consider it more appropriate to graphically represent the relative colony formation after *Kdm3a/b* knockdown, normalized to the non-targeting control, in both conditions (dm- α KG and DMSO, respectively).

2. The color codes may be inverted in the graphs in Fig. 3C, Fig.3H right panel.

We have corrected this error.

3. Fig.3G left panel; EpiLCs in the legend may be ESCs w/o 2i.

We have amended the legend accordingly.

4. page 9, 3rd paragraph, lines 1- 2; 'Addition of dm-aKG up to 24h after-----' may be 'Addition of dm-aKG from 24h to 48h after-----'.

We have modified the sentence as suggested.

5. page 12, lines 2-3; Please cite a reference or show data concerning OSN binding in *Idh2*.

We now show data on OSN binding within a region of 20Kb upstream to 4Kb downstream of the transcriptional start site of *Idh2*; by contrast, we do not find this to be the case for *Idh1*, and *Dist*. Please see graphical representations (see Appendix Figure S1), and our reference to it in the text (see page 12, last paragraph, lines 14-15, continued on page 13, lines 1-3). We provide details on evaluating OSN binding, including the three references to the CHIP-seq data sets we have analyzed, in the Methods section (see page 25, last paragraph, continued on page 26).

Referee #3:

This is an excellent and novel manuscript that for the first time systematically tests the effect of aKG co-factor on distinct states of pluripotency, during their transition and in PGCs. This is a much needed study and set of results following the initial study by Carey et al. Nature 2015 showing that Glutamine helps pluripotency via aKG production, however since then systematic characterization of aKG on pluripotency was missing (and in different states and during their transitions).

The authors use exacting reporters and cutting edge differentiation protocols (e.g. REX1-dGFP, Blimp1-mVenus and PGC reporters) to test the result and responsive genes to aKG. They establish beyond any doubt that aKG supports naive and PGC state and not the primed states. They do not stop there, but also show the importance of TET enzymes and KDM members to directly dictate the epigenetic state. They also establish difference in TCA cycle gene expression during naive to primed state, that supports the rationale for having aKG levels high in naive state.

The manuscript is very well written, and all conclusions are strongly supported by the results. I have no meaningful comments to add or experiments to request, and support publishing this exciting and thorough study.

We acknowledge and appreciate the supportive comments by the Reviewer.

References

Hackett JA, Sengupta R, Zylicz JJ, Murakami K, Lee C, Down TA, Surani MA (2013) Germline DNA Demethylation Dynamics and Imprint Erasure Through 5-Hydroxymethylcytosine. *Science* 339: 448-452

Kurimoto K, Yabuta Y, Hayashi K, Ohta H, Kiyonari H, Mitani T, Moritoki Y, Kohri K, Kimura H, Yamamoto T, Katou Y, Shirahige K, Saitou M (2015) Quantitative Dynamics of Chromatin Remodeling during Germ Cell Specification from Mouse Embryonic Stem Cells. *Cell Stem Cell* 16: 517-532

Pusapati R, Settleman J (2016) TORquing metabolic reprogramming in cancer cells. *Cell Cycle* 15: 2387-2388

Sottnik JL, Lori JC, Rose BJ, Thamm DH (2011) Glycolysis inhibition by 2-deoxy-d-glucose reverts the metastatic phenotype in vitro and in vivo. *Clin Exp Metastas* 28: 865-875

Zhou W, Choi M, Margineantu D, Margaretha L, Hesson J, Cavanaugh C, Blau CA, Horwitz MS, Hockenbery D, Ware C, Ruohola-Baker H (2012) HIF1alpha induced switch from bivalent to exclusively glycolytic metabolism during ESC-to-EpiSC/hESC transition. *EMBO J* 31: 2103-16

Zylicz JJ, Dietmann S, Gunesdogan U, Hackett JA, Cougot D, Lee C, Surani MA (2015) Chromatin dynamics and the role of G9a in gene regulation and enhancer silencing during early mouse development. *Elife* 4

Figure 1. Addition of dm-αKG at t=24h to t=48h during the ESC-to-EpiLC transition retains a *Rex1*-high state. Shown is the fraction of *Rex1*-GFPd2 cells, following dm-αKG supplementation from t=24h, averaged over duplicate experiments. Error bars indicate \pm SE.

Thank you for submitting your revised manuscript for consideration by The EMBO Journal. Your revised study was sent back to two of the original referees for re-evaluation, and we have received comments from both of them, which I enclose below. As you will see the referees find that their concerns have been sufficiently addressed and they are now broadly favour of publication.

Thus, we are pleased to inform you that your manuscript has been accepted in principle for publication in The EMBO Journal, pending some minor issues regarding formatting of the study as outlined below, which need to be adjusted at re-submission.

REFEREE COMMENTS:

Referee #1:

My concerns have been adequately addressed. This is a very topical paper.

Referee #2:

All of my concerns have been appropriately addressed in the revised manuscript.

Corresponding Author Name: M Azim Surani

Manuscript Number: EMBOJ-2018-99518